# PSMix: Robust Point Cloud Recognition through Spectral Domain Mixing

**Xin Wei** [1]   **Qin Yang** [1]   **Hongji Zhao** [1]   **Fei Gao** [1]   **Mingrui Zhu** [1]   **Nannan Wang** [1]   **Xinbo Gao** [1]

## Abstract

While data augmentation is essential for robust point cloud recognition, conventional spatial mixup strategies often compromise geometric integrity by generating physically unrealistic samples. To overcome this limitation, we propose PSMix, which shifts the mixing paradigm to the spectral domain via the Spherical Harmonic Transform. Instead of simple coordinate interpolation, PSMix performs a rotation-aware hierarchical mixing on spectral coefficients. This approach explicitly preserves global structural properties while diversifying local details, achieving a balance that spatial methods struggle to maintain. Complementing this, we introduce an adversarial rotation optimization strategy to enforce invariance against challenging orientations. Extensive experiments on ModelNet-C and ScanObjectNN-C demonstrate that PSMix achieves state-of-the-art robustness, while also serving as an orthogonal plug-in that further boosts the performance of existing spatial strategies. Code will be made available at https://github.com/qinyxdu/PSMix.

## 1. Introduction

Point clouds serve as a fundamental data format in 3D computer vision, supporting applications ranging from autonomous driving to embodied AI. While deep learning models (Qi et al., 2017; Wang et al., 2019; Xiang et al., 2021; Tan et al., 2024) have achieved high accuracy on clean benchmarks, their performance often degrades significantly under out-of-distribution (OOD) conditions and real-world data corruptions. This sensitivity highlights the necessity of developing robust representations for practical deployments.

Data augmentation, particularly Mixup-based strategies (Zhang et al., 2018; Yun et al., 2019; Uddin et al.; Kim et al.; Chen et al., 2023; Qin et al., 2024), has proven

[1]Xidian University. Correspondence to: Nannan Wang <nnwang@xidian.edu.cn>.

*Proceedings of the 43$^{rd}$ International Conference on Machine Learning*, Seoul, South Korea. PMLR 306, 2026. Copyright 2026 by the author(s).

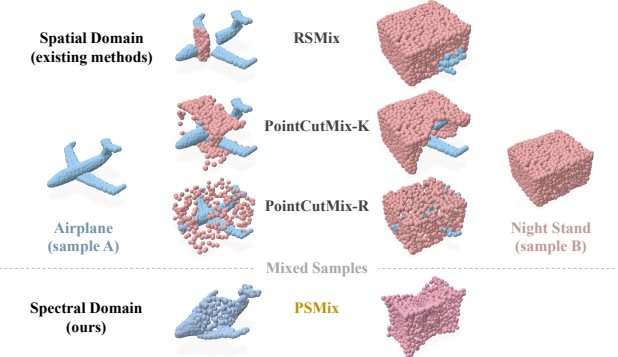

*Figure 1.* Comparison of spatial and spectral mixup of point clouds. While spatial methods (top three rows) introduce geometric discontinuities, PSMix (bottom row) leverages spectral interpolation to generate smooth, structurally coherent augmented samples.

effective in enhancing robustness by regularizing decision boundaries. However, extending Mixup to 3D point clouds presents distinct challenges due to the irregular and unordered nature of the data. Existing methods typically operate in the spatial domain, relying on coordinate interpolation or patch replacement. Although these approaches increase data diversity, they often disrupt geometric integrity. As illustrated in Figure 1, spatial mixing can generate samples characterized by sharp discontinuities and fragmented regions. Such artifacts distort the global topology of objects, potentially hindering the model from learning generalized geometric features.

To overcome these spatial limitations, we shift the augmentation paradigm to the spectral domain, which naturally captures global geometric properties of point clouds (Hu et al., 2022; Liu et al., 2023; Han et al., 2023; Wei et al., 2025). In this vein, we propose PSMix, a framework that utilizes the Spherical Harmonic Transform (SHT) to treat point clouds as continuous spherical functions. Specifically, we first employ Wigner D-matrices to perform rotation directly on the spectral coefficients. Critically, to enforce invariance against orientation shifts, we integrate an Adversarial Rotation Optimization strategy. This mechanism dynamically identifies the most challenging rotation parameters through a minimax objective, compelling the model to learn robust features against severe geometric transformations. Following this, we apply a rotation-aware hierarchical mixing strategy that preserves the global structural topol-

ogy encoded in low-frequency coefficients while integrating local geometric details from high-frequency components. Finally, the augmented spectral representation is mapped back to the coordinate space via the Inverse SHT, ensuring that the generated samples maintain surface continuity and structural coherence.

Extensive experiments on ModelNet-C and ScanObjectNN-C validate the efficacy of our approach. PSMix achieves state-of-the-art robustness across multiple backbones and serves as an orthogonal plug-in to existing spatial augmentations. When combined with spatial methods, PSMix yields further performance gains, demonstrating its versatility in a joint spatial-spectral framework.

Our contributions are summarized as follows:

(1) We propose PSMix, a spectral augmentation framework that generates topologically consistent point clouds, addressing the discontinuity issues found in spatial mixup methods.
(2) We introduce a rotation-aware hierarchical fusion mechanism for SHT coefficients, enhanced by adversarial optimization to improve invariance to orientation shifts.
(3) We demonstrate that PSMix achieves state-of-the-art robustness on standard benchmarks and exhibits strong complementarity with existing augmentation techniques.

## 2. Related Works

### 2.1. Data Augmentation for Point Clouds

Data augmentation is a fundamental strategy for improving the generalization and robustness of 3D deep learning models. While early approaches focused on global affine transformations of point clouds such as random rotation, scaling, and jittering (Qi et al., 2017; Wang et al., 2019), recent research has adapted mixing-based strategies to the 3D domain, which were originally proposed for image classification (Zhang et al., 2018).

Most existing methods operate directly in the spatial domain. PointMixup (Chen et al., 2020) interpolates optimized assignment between point clouds. RSMix (Lee et al., 2021) and PointCutMix (Zhang et al., 2022) introduce part-level swapping to preserve local structures. PointPatchMix (Wang et al., 2024) further utilizes patch significance scores for mixing. While these methods increase data diversity, they rely on Euclidean space operations that often disrupt the intrinsic topological consistency of 3D objects. As shown in Figure 1, spatial interpolation can create physically implausible artifacts and discontinuous boundaries, limiting the model's robustness against real-world corruptions.

### 2.2. Spectral Analysis for Point Clouds

Spectral methods provide a robust alternative to spatial approaches by decomposing geometric shapes into fre-

quency components. One prominent direction involves the Graph Fourier Transform (GFT), which utilizes the eigen-decomposition of the graph Laplacian. While effective for single-object tasks like denoising (Zhang et al., 2020), adversarial attacks (Liu et al., 2023) and test-time adaptation (Hu et al., 2022), GFT generates basis vectors that are instance-specific. This dependency creates a basis mismatch between different objects, rendering GFT mathematically ill-posed for cross-sample tasks like Mixup, as coefficients from disparate graphs cannot be directly combined.

To address this limitation, some works have leveraged the Spherical Harmonics Transform (SHT) (Kazhdan et al., 2003; Saupe & Vranić, 2001), which defines a canonical basis on the unit sphere. Unlike GFT, SHT provides a universal spectral domain, ensuring that coefficients from different point clouds are aligned and mathematically comparable. This property is fundamental for spectral mixing strategies. Furthermore, SHT offers desirable geometric properties for augmentation; specifically, it enables rotation-equivariant transformations via Wigner D-matrices (Kostelec & Rockmore, 2008), avoiding the discontinuities associated with Euler angles or quaternions. Although SHT has been applied to shape retrieval (Poulenard et al., 2019) and equivariant learning (Esteves et al., 2018; Lee & Cho, 2024), its potential for generating structurally consistent samples via hierarchical mixing remains underexplored.

## 3. Method

In this section, we present PSMix, a spectral-domain data augmentation framework designed to enhance the robustness of 3D point cloud recognition. The overall pipeline is illustrated in Figure 2. Specifically, given initial sample pairs $(P, Y_P)$ and $(Q, Y_Q)$, we first transform the point clouds into their respective spectral representations, $\hat{P}$ and $\hat{Q}$, via the Spherical Harmonics Transform (SHT). Next, the coefficients $\hat{Q}$ undergo an optimized spectral rotation using Wigner D-matrices parameterized by $R(\alpha, \beta, \gamma)$, yielding the rotated coefficients $\hat{Q}_{rot}$. Subsequently, $\hat{P}$ and $\hat{Q}_{rot}$ are hierarchically mixed to generate the combined spectral coefficients $\hat{P}_{mix}$. Finally, $\hat{P}_{mix}$ is transformed back to the spatial domain through the Inverse SHT to produce the augmented point cloud $\tilde{P}$. Concurrently, the corresponding augmented label $\tilde{Y}$ is generated by mixing the original labels $Y_P$ and $Y_Q$.

### 3.1. Preliminaries: Spherical Harmonics Transform

To address the instance-specific basis limitation of the Graph Fourier Transform (GFT) in inter-cloud mixing tasks, we employ the Spherical Harmonics Transform (SHT) (Kazhdan et al., 2003; Saupe & Vranić, 2001; Venkatraman et al., 2009; Ducroz et al., 2012; Poulenard et al., 2019; Lee & Cho, 2024). SHT generalizes the Fourier series to functions defined on the

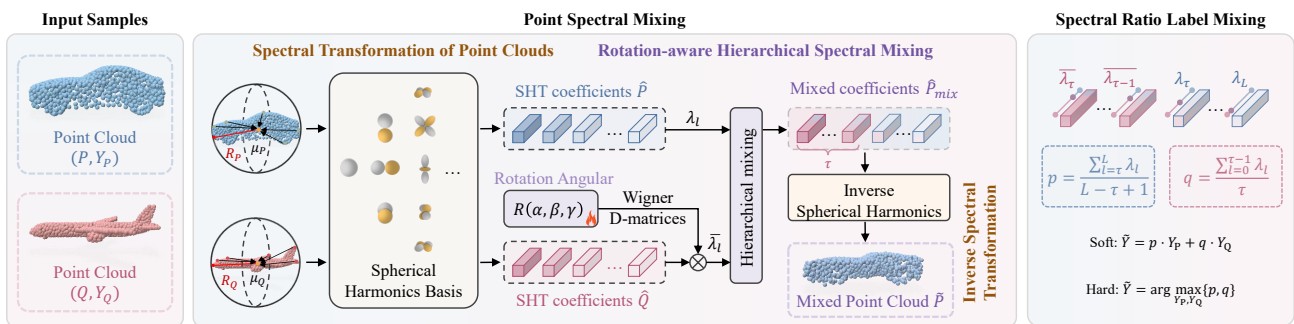

**Figure 2. Architecture of PSMix.** The framework fuses inputs $(P, Y_P)$ and $(Q, Y_Q)$ via two streams: (1) **Point Spectral Mixing** transforms point clouds into SHT coefficients, performs rotation-aware hierarchical mixing, and reconstructs $\tilde{P}$ via Inverse SHT. (2) **Spectral Ratio Label Mixing** concurrently generates the target label $\tilde{Y}$ using soft or hard strategies based on spectral energy distribution.

unit sphere $\mathbb{S}^2 \subset \mathbb{R}^3$, providing an orthogonal and complete basis formed by the spherical harmonics $Y_l^m(\theta, \phi)$:

$$Y_l^m(\theta, \phi) = N_l^m P_l^m(\cos \phi) e^{im\theta}, \qquad (1)$$

where $\phi \in [0, \pi]$ denotes the polar angle, $\theta \in [0, 2\pi]$ is the azimuthal angle, degree $l \in \mathbb{N}$, order $m \in \mathbb{Z}$ with $|m| \le l$, and $i = \sqrt{-1}$. The normalization coefficient $N_l^m$ is defined as:

$$N_l^m = (-1)^m \sqrt{\frac{2l+1}{4\pi} \frac{(l-m)!}{(l+m)!}}, \qquad (2)$$

and $P_l^m(\cdot)$ represents the associated Legendre function:

$$P_l^m(x) = (-1)^m (1 - x^2)^{\frac{m}{2}} \frac{d^m}{dx^m} P_l(x), \qquad (3)$$

where $P_l(x)$ is the Legendre polynomial of degree $l$. Since spherical harmonics form a complete orthogonal basis for square-integrable functions on $\mathbb{S}^2$, any function $f : \mathbb{S}^2 \to \mathbb{C}$ can be expanded as:

$$f(\theta, \phi) = \sum_{l=0}^{\infty} \sum_{m=-l}^{l} c_l^m Y_l^m(\theta, \phi), \qquad (4)$$

where the SHT coefficients $c_l^m$ are computed via:

$$c_l^m = \int_0^\pi \int_0^{2\pi} f(\theta, \phi) \overline{Y_l^m(\theta, \phi)} \sin \phi \, d\theta \, d\phi, \qquad (5)$$

with $\overline{Y_l^m}$ denoting the complex conjugate.

SHT offers two fundamental advantages over GFT for point cloud augmentation. First, SHT provides a **universal, canonical basis** defined on the unit sphere. Unlike GFT, where bases are dependent on the specific graph topology of each object, SHT projects all point clouds onto a shared spectral space. This universality enables the mathematically valid linear combination of spectral coefficients from disparate objects. Second, SHT coefficients are **rotation-equivariant**.

A spatial rotation of a 3D object induces a predictable linear transformation of its spectral coefficients via Wigner D-matrices (Kostelec & Rockmore, 2008). This property is critical for optimization, as it avoids the discontinuities and singularities (*e.g.*, gimbal lock) associated with traditional spatial representations like Euler angles (Lee & Cho, 2024; Zhou et al., 2019). By combining a unified spectral basis for fusion with optimization-friendly rotation, SHT serves as an ideal mathematical foundation for the proposed PSMix framework.

### 3.2. Point Spectral Mixing

#### 3.2.1. Spectral Transformation of Point Clouds

The first stage of PSMix projects the discrete input point clouds $P$ and $Q$, taken from samples $(P, Y_P)$ and $(Q, Y_Q)$, into a continuous spectral domain.

**Normalization.** As spherical harmonics are defined on the unit sphere $\mathbb{S}^2$, we first normalize the input point clouds to fit within this domain. The normalized point clouds $P'$ and $Q'$ are computed as:

$$P' = \frac{P - \mu_P}{R_P}, \quad Q' = \frac{Q - \mu_Q}{R_Q}, \qquad (6)$$

where $\mu_P, \mu_Q$ represent the centroids, and $R_P, R_Q$ denote the maximum Euclidean distance from the centroid to any point in $P$ and $Q$, respectively.

**Spherical Harmonic Decomposition.** To transition from discrete points to the spectral domain, we convert $P'$ and $Q'$ into continuous spherical functions via density estimation on a spherical grid. This representation effectively captures the global geometric structure needed for robust classification (Poulenard et al., 2019; Esteves et al., 2018). These continuous functions are then decomposed via the Spherical Harmonic Transform (SHT), denoted as $\phi_{\text{SHT}}$, up to a bandwidth (maximum degree) $L$:

$$\hat{P} = \phi_{\text{SHT}}(P'), \quad \hat{Q} = \phi_{\text{SHT}}(Q'). \qquad (7)$$

The resulting spectral representation $\hat{P}$ resides in the complex space $\mathbb{C}^{(L+1)^2}$, comprising all harmonic coefficients $c_l^m$ for degrees $0 \le l \le L$ and orders $-l \le m \le l$.

To facilitate the hierarchical mixing strategy introduced in the subsequent section, we structure these coefficients by degree. Specifically, $\hat{P}$ is formulated as a sequence of degree-specific vectors $\{c_{p,l}\}_{l=0}^L$, where each vector $c_{p,l} \in \mathbb{C}^{2l+1}$ represents the spectral information at frequency level $l$. A corresponding structure, $\{c_{q,l}\}_{l=0}^L$, is derived for $\hat{Q}$. The bandwidth $L$ determines the spectral resolution, where lower degrees correspond to global shape and higher degrees capture local geometric details.

### 3.2.2. Rotation-aware Hierarchical Spectral Mixing

Having obtained the spectral coefficients $\hat{P}$ and $\hat{Q}$, we proceed to fuse them. Direct uniform interpolation, typical in spatial Mixup, is suboptimal here due to the hierarchical nature of SHT coefficients. As shown in Figure 3 (a), low-degree coefficients (small $l$) encode the global shape, while high-degree coefficients (large $l$) capture fine-grained details. A uniform mixing strategy disregards this frequency-dependent structure, potentially resulting in topologically inconsistent samples. To address this, we propose a rotation-aware hierarchical strategy that respects geometric significance across scales while enhancing diversity through spectral rotation.

**Spectral Rotation Augmentation.** To augment sample diversity, we perform rotation directly in the spectral domain on the source coefficients $\hat{Q}$. As illustrated in Figure 3 (b), this operation fundamentally alters the relative alignment of the constituent shapes, creating diverse geometric combinations that are unattainable through fixed-pose mixing. Specifically, for a rotation $R(\alpha, \beta, \gamma) \in SO(3)$ parameterized by Euler angles, each degree-specific coefficient vector $c_{q,l}$ from $\hat{Q} = \{c_{q,l}\}_{l=0}^L$ is transformed via the Wigner $D$-matrix $D^l(R)$:

$$\hat{c}_{q,l} = D^l(R)c_{q,l}. \tag{8}$$

Here, $D^l(R)$ is the $(2l+1) \times (2l+1)$ irreducible representation of the rotation group for degree $l$. This operation yields the rotated spectral coefficients $\hat{Q}_{rot} = \{\hat{c}_{q,l}\}_{l=0}^L$ without introducing spatial interpolation artifacts.

**Hierarchical Coefficients Mixing.** We then mix the target coefficients $\hat{P}$ with the rotated source $\hat{Q}_{rot}$ using a frequency-adaptive mechanism. Leveraging the multi-resolution property of SHT, we define the mixed coefficients $c_{mix,l}$ for each degree $l$ as:

$$c_{mix,l} = \lambda_l \cdot c_{p,l} + (1 - \lambda_l) \cdot \hat{c}_{q,l}, \tag{9}$$

where the mixing ratio $\lambda_l$ varies with frequency to ensure geometric coherence. We model $\lambda_l$ using a sigmoid function

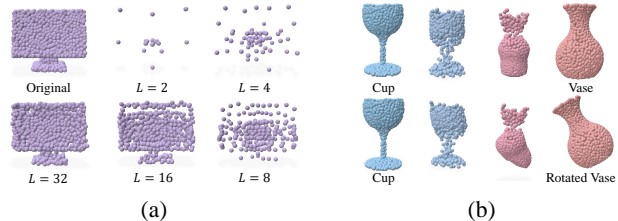

(a)                      (b)

*Figure 3.* Visual analysis of SHT properties. (a) **Hierarchical Reconstruction**: Increasing the degree $L$ progressively recovers fine-grained details from the global shape. (b) **Spectral Rotation**: Mixing a cup with a fixed vase (top) *vs.* a spectrally rotated vase (bottom). The proposed spectral rotation introduces diverse relative orientations, significantly enriching geometric variability.

centered at a transition bandwidth $\tau$:

$$\lambda_l = \sigma(l - \tau). \tag{10}$$

For low frequencies ($l < \tau$), $\lambda_l \to 0$, causing $c_{mix,l}$ to be dominated by $\hat{c}_{q,l}$ (inheriting the global structure of $Q$). Conversely, for high frequencies ($l > \tau$), $\lambda_l \to 1$, ensuring $c_{mix,l}$ retains the fine details of $P$ (via $c_{p,l}$). This strategy synthesizes a structurally coherent hybrid sample $\hat{P}_{mix} = \{c_{mix,l}\}_{l=0}^L$ that preserves the global topology of the source object while integrating texture-like details from the target.

### 3.2.3. Inverse Spectral Transformation

Finally, we reconstruct the spatial representation from the fused spectral coefficients $\hat{P}_{mix}$. The Inverse Spherical Harmonic Transform (ISHT), denoted as $\phi_{\text{ISHT}}$, maps the spectral data back to the coordinate domain to yield the augmented point cloud $\tilde{P}$:

$$\tilde{P} = \phi_{\text{ISHT}}(\hat{P}_{mix}). \tag{11}$$

### 3.3. Spectral Ratio Label Mixing

Unlike spatial Mixup which applies a uniform ratio across all features, PSMix modulates spectral components hierarchically. Consequently, the label generation must reflect the varying contributions of the source point clouds across different frequency bands. To achieve this, we introduce two labeling strategies: PSMix-S (Soft) and PSMix-H (Hard).

**PSMix-S.** This variant employs a soft labeling mechanism based on the spectral energy distribution. We derive the aggregate mixing weights, $p$ and $q$, by averaging the degree-specific mixing ratios $\lambda_l$ (defined in Eqn. 10) across the high-frequency and low-frequency bands, respectively:

$$p = \frac{1}{L - \tau + 1} \sum_{l=\tau}^{L} \lambda_l, \quad q = \frac{1}{\tau} \sum_{l=0}^{\tau-1} \lambda_l. \tag{12}$$

Since we typically select a small transition threshold $\tau$, the low-frequency components (from $Q$) primarily encode the

coarse envelope. In contrast, semantic identity is largely determined by the rich geometric details preserved in the high-frequency components (from $P$). Consequently, Eq. 12 naturally assigns a higher weight to $p$, ensuring the augmented label aligns with the source contributing the dominant discriminative features. The final augmented label $\tilde{Y}$ is then computed as a weighted combination:

$$\tilde{Y} = p \cdot Y_P + q \cdot Y_Q. \quad (13)$$

**PSMix-H.** This variant assigns a deterministic hard label by identifying the dominant spectral contributor. The label is selected based on the maximum weight between $p$ and $q$:

$$\tilde{Y} = \underset{Y_P, Y_Q}{\arg\max}\{p, q\}. \quad (14)$$

By producing a hard label, PSMix-H effectively functions as a single-sample augmentation strategy (treating the mixed sample as a perturbed version of the dominant class). This property allows it to be seamlessly integrated with existing spatial domain mixing methods, establishing a **Joint Spatial-Spectral Data Augmentation** framework. In our experiments, we demonstrate this orthogonality by combining PSMix-H with PointMixup (Chen et al., 2020), RSMix (Lee et al., 2021), and variants of PointCutMix (Zhang et al., 2022). As detailed in Table 2, this joint approach yields cumulative gains in model robustness.

### 3.4. Training Strategy

3.4.1. ADVERSARIAL ROTATION OPTIMIZATION

To improve model robustness against diverse orientations, we treat the rotation parameters $R$ in the Wigner D-matrices (Eqn. 8) as learnable variables. We propose an adversarial optimization strategy that seeks a base rotation $R_0$, augmented by a stochastic sampling mechanism to promote training stability.

During training, the rotation parameters $R$ applied to the spectral coefficients $\hat{Q}$ are sampled from a Gaussian distribution $\mathcal{N}(R_0, \delta)$, centered at the learnable base rotation $R_0$. To facilitate convergence, the variance $\delta$ of this distribution is linearly decayed as training progresses. The base rotation $R_0$ is updated via a minimax optimization objective:

$$\min_{\theta} \max_{R_0} \mathcal{L}_{cls}(f_\theta(\tilde{P}(R_0)), \tilde{Y}), \quad (15)$$

where $f_\theta$ denotes the classification network parameterized by $\theta$, and $\tilde{P}(R_0)$ represents the augmented point cloud generated using the base rotation $R_0$.

This framework establishes a minimax objective: the inner maximization searches for the rotation $R_0$ that increases the classification loss, aiming to generate challenging geometric orientations. Concurrently, the outer minimization updates

the network parameters $\theta$ to minimize the loss on these perturbed samples. We employ an alternating optimization schedule to update $R_0$ and $\theta$ iteratively. The stochastic sampling around $R_0$ is designed to smooth the optimization process, helping to mitigate the risk of collapsing into trivial local solutions.

3.4.2. OBJECTIVE FUNCTION

Following the strategy in PointCutMix (Zhang et al., 2022), we introduce a hyperparameter $\rho \in [0, 1]$ to control the augmentation probability. Specifically, the application of PSMix is governed by a Bernoulli distribution with parameter $\rho$. The binary indicator $\mathbb{I}_\rho$ determines the data flow: when $\mathbb{I}_\rho = 0$, the model is optimized on the original input; when $\mathbb{I}_\rho = 1$, spectral augmentation is applied. The total training objective $\mathcal{L}$ is formulated as:

$$\mathcal{L} = (1 - \mathbb{I}_\rho) \cdot \mathcal{L}_{cls}(f_\theta(P), Y_P) + \mathbb{I}_\rho \cdot \mathcal{L}_{cls}(f_\theta(\tilde{P}), \tilde{Y}), \quad (16)$$

where $\mathcal{L}_{cls}$ denotes the classification loss. This formulation enables the model to integrate information from both clean data distributions and augmented samples, facilitating a balanced training paradigm.

## 4. Experiments

**Datasets.** We evaluate the robustness on two standard benchmarks: **ModelNet40-C** and **ScanObjectNN-C**. ModelNet40-C is generated from the synthetic ModelNet40 (Wu et al., 2015) test set (2,468 shapes across 40 categories), whereas ScanObjectNN-C is derived from the real-world ScanObjectNN (Uy et al., 2019) test set (581 samples across 15 classes). Both benchmarks adhere to a unified protocol comprising 15 corruption types organized into three categories: (1) **Density**: Density Decrease (DD), Density Increase (DI), Cutout (CO), Occlusion (OC), and LiDAR (LD); (2) **Noise**: Uniform (UNI), Gaussian (GAU), Background (BG), Impulse (IP), and Upsampling (US); and (3) **Transformation**: Radial Basis Function (RBF), Inverse RBF (IR), Shear (SH), Rotation (ROT), and Free-Form Deformation (FFD).

**Backbones and Comparable Methods.** We select the supervised methods DGCNN (Wang et al., 2019) and PointNeXt (Qian et al., 2022) as the base classifier, denoted $f_\theta$, due to their strong performance on the ModelNet40 and ScanObjectNN datasets. Additionally, we evaluate PSMix on Point-MAE (Pang et al., 2022), a more sophisticated, transformer-based, self-supervised model. To evaluate PSMix-H/S against existing 3D domain augmentation methods, we implement seven comparable approaches using their publicly available open-source code: PointMixup (Chen et al., 2020), RSMix (Lee et al., 2021), PGD (Sun et al., 2021a), PointCutMix (Zhang et al., 2022) (with its -K and -R variants), AdaptPoint (Wang et al., 2023), and Point-

*Table 1.* Classification accuracy (%) on ModelNet40-C. Models are trained on clean ModelNet40. "Vanilla" denotes the baseline without augmentation. **Bold** and underlined indicate the highest and second highest values, respectively.

| | Method | UNI | GAU | BG | IP | US | RBF | IR | DD | DI | SH | ROT | CO | FFD | OC | LD | Avg. | Gain |
|---|---|---|---|---|---|---|---|---|---|---|---|---|---|---|---|---|---|---|
| DGCNN | Vanilla | 79.45 | 73.41 | 61.66 | 61.83 | 67.09 | 77.75 | 80.38 | 72.48 | 83.02 | 84.64 | 59.15 | 75.68 | 80.10 | 33.06 | 11.70 | 66.76 | - |
| | PointMixup | 88.05 | 86.75 | 55.92 | 84.03 | **84.96** | 85.45 | 86.14 | 79.01 | 87.52 | 83.95 | 63.29 | 83.18 | 84.56 | 35.29 | 29.05 | 74.48 | ↑7.72 |
| | RSMix | 84.23 | 79.78 | 87.31 | 88.04 | 78.24 | 80.51 | 81.68 | 88.01 | 89.30 | 86.46 | 59.32 | **89.18** | 83.63 | 40.68 | 34.68 | 76.74 | ↑9.98 |
| | PGD | **89.42** | **89.50** | 15.80 | **88.94** | 82.05 | 86.34 | 87.07 | 74.27 | 85.05 | 80.14 | 56.07 | 78.65 | 82.46 | 34.44 | 33.39 | 70.91 | ↑4.15 |
| | PointCutMix-K | 83.79 | 77.39 | **89.75** | 86.38 | 78.88 | 83.51 | 85.41 | **88.41** | **89.78** | 88.25 | 62.84 | 88.37 | 86.22 | 38.41 | 27.18 | 76.97 | ↑10.21 |
| | PointCutMix-R | 85.74 | 83.75 | 89.71 | 88.61 | 81.07 | 83.87 | 85.05 | 87.44 | 88.98 | 85.57 | 62.19 | 86.99 | 84.11 | 36.99 | 29.41 | 77.30 | ↑10.54 |
| | AdaptPoint | 82.33 | 77.07 | 39.95 | 68.19 | 72.20 | 86.51 | 87.12 | 83.95 | 86.10 | 87.28 | **85.74** | 81.69 | 86.55 | 37.32 | 33.39 | 73.03 | ↑6.27 |
| | **PSMix-H(Ours)** | 87.88 | 86.91 | 70.34 | 86.39 | 84.60 | 88.29 | **88.65** | 81.32 | 87.84 | 88.01 | 73.66 | 82.09 | 87.40 | 39.71 | **35.45** | 77.90 | ↑11.14 |
| | **PSMix-S(Ours)** | 89.06 | 87.56 | 72.20 | 86.18 | 84.93 | 88.41 | 88.29 | 81.16 | 87.24 | 88.65 | 75.77 | 81.56 | 86.63 | 40.15 | 35.29 | 78.21 | ↑**11.45** |
| PointNeXt | Vanilla | 59.81 | 50.20 | 53.93 | 63.37 | 73.38 | 68.64 | 70.54 | 81.40 | 81.89 | 74.84 | 37.52 | 82.33 | 69.08 | 42.87 | 32.90 | 62.85 | - |
| | PointMixup | 86.22 | 84.52 | 63.86 | 84.24 | 88.21 | 85.17 | 86.63 | 86.75 | 87.56 | 82.66 | 54.13 | 87.12 | 83.63 | 42.71 | 34.20 | 75.84 | ↑12.99 |
| | RSMix | 44.89 | 37.44 | 86.87 | **88.49** | 65.84 | 58.10 | 60.70 | 89.14 | 90.88 | 68.52 | 28.32 | **90.60** | 62.16 | 43.60 | 36.63 | 63.48 | ↑0.63 |
| | PGD | **89.26** | **88.49** | 6.73 | 83.67 | **89.22** | 84.60 | 86.75 | 83.06 | 88.41 | 78.73 | 55.35 | 79.62 | 80.39 | 40.48 | 40.56 | 71.69 | ↑8.84 |
| | PointCutMix-K | 62.40 | 53.24 | 89.79 | 79.13 | 81.07 | 73.58 | 75.41 | 86.02 | 87.28 | 75.97 | 38.45 | 88.05 | 75.00 | **46.31** | 36.55 | 69.88 | ↑7.03 |
| | PointCutMix-R | 79.90 | 76.50 | **89.99** | 88.17 | 86.79 | 76.18 | 78.28 | 87.16 | 88.21 | 75.61 | 43.40 | 88.45 | 75.08 | 39.55 | 36.59 | 73.99 | ↑11.14 |
| | AdaptPoint | 81.36 | 78.28 | 39.18 | 60.25 | 82.58 | 86.47 | 86.67 | 83.79 | 85.05 | 87.16 | **84.60** | 82.90 | 86.30 | 42.06 | **43.23** | 73.99 | ↑11.14 |
| | **PSMix-H(Ours)** | 88.57 | 87.12 | 70.79 | 82.66 | 89.02 | **89.55** | **90.40** | 85.37 | 89.34 | 86.30 | 70.71 | 83.79 | 86.43 | 42.50 | 35.33 | 78.53 | ↑15.68 |
| | **PSMix-S(Ours)** | 88.45 | 86.71 | 75.57 | 84.16 | 89.02 | 87.80 | 89.55 | 85.98 | 89.71 | 85.37 | 73.50 | 84.40 | 87.32 | 42.71 | 36.30 | 79.10 | ↑**16.25** |

PatchMix[1] (Wang et al., 2024), which represent two main categories in point cloud augmentation: mix-based augmentation (PointMixup, PointCutMix, RSMix, PointPatchMix) and generative augmentation (PGD, AdaptPoint).

**Implementation Details.** We implement the spherical harmonic transforms using the efficient, GPU-accelerated `torch-harmonics` library (Bonev et al., 2023). To introduce spectral resolution diversity during training, the maximum SHT degree $L$ is uniformly sampled from the integer interval $[15, 70]$, with the hierarchical transition point $\tau$ fixed at 3 for all experiments to maintain a unified setting. We employ an alternating optimization schedule initiated with random Euler angle parameters. These rotation parameters are updated every five training epochs by maximizing the classification loss defined in Eqn. 15. During the backbone update phase, rotations are sampled using the current Euler parameters, where the sampling variance $\delta$ is linearly annealed from $\pi$ to $\pi/36$ over the course of training to stabilize convergence. We train DGCNN and PointNeXt from scratch, and fine-tune Point-MAE using official weights, all for 300 epochs with $1,024$ input points and a batch size of 32. The total objective in Eqn. 16 is minimized using the Adam optimizer (Kingma & Ba, 2015) with an initial learning rate of $10^{-3}$.

### 4.1. Classification Results on ModelNet-C

Table 1 demonstrates the effectiveness of PSMix variants on ModelNet-C across DGCNN and PointNeXt backbones. On DGCNN, PSMix-S achieves a leading mean accuracy

---

[1]We re-implemented PointPatchMix due to the unavailability of the official patch scoring module, evaluating it exclusively on Point-MAE consistent with its transformer-based design.

*Table 2.* Classification accuracy (%) of the Joint Spatial-Spectral Data Augmentation strategy on ModelNet40-C. We compare standard spatial augmentation methods against their counterparts integrated with PSMix-H. Green indicates accuracy improvement relative to the corresponding spatial-only baseline, while red denotes a decrease.

| | Method | Density | Noise | Trans. | Avg. |
|---|---|---|---|---|---|
| DGCNN | Vanilla | 55.19 | 68.69 | 76.40 | 66.76 |
| | PointMixup | 62.81 | 79.94 | 80.68 | 74.48 |
| | +PSMix-H | 65.46 ↑2.65 | 86.74 ↑6.80 | 84.63 ↑3.95 | 78.94 ↑4.46 |
| | RSMix | 68.37 | 83.52 | 78.32 | 76.74 |
| | +PSMix-H | 67.93 ↓0.44 | 87.08 ↑3.56 | 86.45 ↑8.13 | 80.49 ↑3.75 |
| | PointCutMix-K | 66.43 | 83.24 | 81.25 | 76.97 |
| | +PSMix-H | **68.86** ↑2.43 | 88.50 ↑5.26 | **88.03** ↑6.78 | **81.80** ↑4.83 |
| | PointCutMix-R | 65.96 | 85.78 | 80.16 | 77.30 |
| | +PSMix-H | 67.64 ↑1.68 | **88.91** ↑3.13 | 85.94 ↑5.78 | 80.83 ↑3.53 |
| PointNeXt | Vanilla | 64.28 | 60.14 | 64.12 | 62.85 |
| | PointMixup | 67.67 | 81.41 | 78.44 | 75.84 |
| | +PSMix-H | 68.93 ↑1.26 | 85.65 ↑4.24 | 85.06 ↑6.62 | 79.88 ↑4.04 |
| | RSMix | 70.17 | 64.71 | 55.56 | 63.48 |
| | +PSMix-H | **72.91** ↑2.74 | 87.16 ↑22.46 | 81.28 ↑25.72 | 80.45 ↑16.97 |
| | PointCutMix-K | 68.84 | 73.13 | 67.68 | 69.88 |
| | +PSMix-H | 72.30 ↑3.46 | 87.96 ↑14.83 | 85.06 ↑17.38 | **81.77** ↑11.89 |
| | PointCutMix-R | 67.99 | 84.27 | 69.71 | 73.99 |
| | +PSMix-H | 68.91 ↑0.92 | **89.12** ↑4.85 | 84.80 ↑15.09 | 80.94 ↑6.95 |

of 78.21%, closely followed by PSMix-H at 77.90%, both outperforming the leading spatial mix method PointCutMix-R by 0.60% and 0.91%, respectively. Using the PointNeXt backbone, PSMix-S obtains 79.10% mean accuracy and PSMix-H achieves 78.53%, compared to a 62.85% baseline. These results represent improvements of 3.26% for PSMix-S and 2.69% for PSMix-H over PointMixup (75.84%), the second-best spatial mixing method. Notably, PSMix variants exhibit superior robustness against transformation corrup-

*Table 3.* Classification accuracy (%) on ScanObjectNN-C. Models are trained on clean ScanObjectNN. "Vanilla" denotes the baseline without augmentation. **Bold** and underlined indicate the highest and second highest values, respectively.

| | Method | UNI | GAU | BG | IP | US | RBF | IR | DD | DI | SH | ROT | CO | FFD | OC | LD | Avg. | Gain |
|---|---|---|---|---|---|---|---|---|---|---|---|---|---|---|---|---|---|---|
| DGCNN | Vanilla | 40.45 | 35.97 | 66.95 | 69.02 | 41.82 | 68.50 | 69.71 | 62.65 | 68.33 | 70.22 | 65.23 | 65.58 | 70.57 | 8.43 | 9.64 | 54.20 | - |
| | PointMixup | 50.60 | 44.41 | 63.51 | 66.27 | 45.09 | 69.54 | 69.19 | 65.40 | 67.47 | 71.26 | 62.13 | 66.78 | 68.67 | 9.81 | 11.02 | 55.41 | ↑1.21 |
| | RSMix | 42.17 | 39.24 | 73.15 | **73.84** | 37.52 | 68.67 | 68.16 | 66.95 | 66.61 | 71.60 | 63.17 | 65.40 | 69.71 | 10.50 | 9.29 | 55.08 | ↑0.88 |
| | PGD | **55.77** | **56.28** | 14.29 | 53.70 | 51.64 | 55.94 | 56.80 | 40.96 | 52.50 | 55.42 | 51.64 | 50.77 | 55.77 | 10.33 | 8.43 | 44.68 | ↓9.52 |
| | PointCutMix-K | 37.01 | 34.08 | **80.72** | **73.84** | 41.14 | 68.16 | 67.64 | 64.20 | 66.95 | 70.22 | 63.17 | 66.78 | 68.67 | 8.78 | 9.47 | 54.72 | ↑0.52 |
| | PointCutMix-R | 38.04 | 35.63 | 79.52 | 72.46 | 44.92 | 66.44 | 69.02 | 65.06 | 68.50 | 69.54 | 61.27 | 66.09 | 69.54 | 8.78 | 11.02 | 55.05 | ↑0.85 |
| | AdaptPoint | 47.16 | 40.28 | 31.50 | 50.95 | 50.60 | **77.11** | **77.62** | **76.94** | **78.66** | **79.00** | **74.35** | 73.67 | **78.66** | **14.11** | 11.53 | 57.48 | ↑3.28 |
| | **PSMix-H(Ours)** | 50.60 | 48.36 | 61.79 | 73.15 | **61.96** | 76.42 | 76.08 | 73.32 | 77.28 | 76.25 | 72.46 | **74.53** | 77.97 | 12.05 | **11.70** | 61.59 | ↑7.39 |
| | **PSMix-S(Ours)** | 55.59 | 51.12 | 65.23 | 73.32 | 61.45 | 76.42 | **77.62** | 75.56 | 75.73 | 76.42 | 72.46 | 72.46 | 76.59 | 11.70 | 11.19 | **62.19** | ↑7.99 |
| PointNeXt | Vanilla | 20.48 | 39.59 | 58.52 | 27.02 | 27.54 | 49.91 | 51.29 | 83.65 | 61.27 | 48.19 | 44.23 | 83.65 | 53.01 | 8.43 | 8.26 | 44.34 | - |
| | PointMixup | 34.08 | 64.03 | 69.02 | 37.18 | 38.04 | 45.96 | 45.09 | 83.48 | 70.40 | 44.23 | 38.55 | 83.65 | 49.05 | 8.43 | 9.12 | 48.02 | ↑3.68 |
| | RSMix | 19.79 | 42.51 | **86.92** | 47.33 | 27.54 | 44.41 | 46.30 | **86.06** | 79.00 | 44.06 | 33.73 | **86.23** | 45.78 | 7.40 | 8.26 | 47.02 | ↑2.68 |
| | PGD | 44.23 | **77.80** | 21.51 | 31.84 | 45.78 | 44.23 | 45.61 | 77.62 | 78.14 | 43.03 | 38.38 | 77.11 | 43.72 | 10.33 | 10.15 | 45.97 | ↑1.63 |
| | PointCutMix-K | 25.47 | 50.26 | 83.99 | 33.56 | 29.26 | 47.33 | 49.40 | 84.68 | 69.88 | 47.68 | 41.65 | 83.82 | 48.36 | 9.12 | 8.09 | 47.50 | ↑3.16 |
| | PointCutMix-R | 27.71 | 54.91 | 85.20 | 46.64 | 34.25 | 49.57 | 52.50 | 85.54 | 70.40 | 49.23 | 42.69 | 83.99 | 51.81 | 9.47 | 8.09 | 50.13 | ↑5.79 |
| | AdaptPoint | 40.10 | 32.70 | 26.51 | 32.53 | 48.88 | **77.62** | 79.00 | 79.86 | **82.10** | 78.31 | 72.63 | 79.69 | 77.11 | **12.91** | **12.05** | 55.47 | ↑11.13 |
| | **PSMix-H(Ours)** | 42.34 | 37.18 | 40.62 | 51.29 | 51.64 | 76.25 | **79.17** | 77.97 | 78.66 | **78.66** | 72.12 | 76.25 | **77.62** | 9.64 | 10.15 | 57.30 | ↑12.96 |
| | **PSMix-S(Ours)** | **47.33** | 40.62 | 37.35 | **51.46** | **56.63** | 76.59 | 76.59 | 76.94 | 77.62 | 76.76 | 70.74 | 76.42 | 75.56 | 11.36 | 10.15 | **57.47** | ↑13.13 |

tions like RBF, IR, and FFD. Using PointNeXt, PSMix-H achieves accuracy of 89.55%, 90.40%, and 86.43% respectively under these corruptions, while PSMix-S attains 87.80%, 89.55%, and 87.32% in the same settings. This robustness is attributed to its spectral augmentation strategy, which increases the diversity of augmented samples.

Moreover, Table 2 evaluates the joint spatial-spectral strategy introduced in Sect. 3.3. Integrating PSMix-H with spatial augmentations yields consistent gains over spatial-only baselines. For DGCNN, this joint approach improves mean accuracy by 1.58%, 4.69%, and 6.16% for density, noise, and transformation corruptions, respectively. The gain of 6.16% against transformation corruptions particularly highlights spectral mixing's effectiveness for enhancing deformation robustness. Similarly, on the PointNeXt backbone, the joint strategy improves overall mean accuracy by 4.04%, 16.97%, 11.89%, and 6.94%, respectively, relative to spatial-only augmentation baselines. These results confirm that PSMix-H offers complementary benefits to spatial methods, establishing the joint framework as essential for comprehensive model robustness.

### 4.2. Classification Results on ScanobjectNN-C

To further assess the robustness of our proposed PSMix-H and PSMix-S methods, we conduct experiments on the challenging ScanObjectNN-C benchmark. Standard baseline models show a pronounced decrease in classification accuracy on ScanObjectNN-C compared to clean data, with DGCNN at 54.20% and PointNeXt at 44.34%. For the DGCNN backbone, PSMix-S achieves a mean accuracy of 62.19%, and PSMix-H gets 61.59%, which are 4.71% and 4.11% higher respectively than AdaptPoint (57.48%). When

applied to the PointNeXt backbone, PSMix-S yields 57.47% mean accuracy and PSMix-H attains 57.30%, outperforming the next best performing method AdaptPoint's 55.47% by 2.00% and 1.83% respectively. On ScanObjectNN-C, PSMix variants frequently achieve top ranks, indicating their broad effectiveness in enhancing model resilience on this difficult dataset.

Consistent with Table 1, PSMix exhibits the strongest gains on transformation corruptions such as RBF, IR, FFD, and ROT, which involve geometric deformations. This is because spectral mixing preserves global geometric consistency across the point cloud, making it more robust to structural distortions, whereas density- and noise-related corruptions (DD, DI, CO, OC, BG) tend to be better addressed by spatial-domain augmentations, as these corruptions primarily introduce noise or background artifacts rather than structural changes. Overall, these observations highlight the complementary strengths of spectral and spatial mixing and clarify the scenarios where each approach is most effective.

*Table 4.* Accuracy (%) and training time (*s*/epoch) on ModelNet40-C with Point-MAE. $T_5$ denotes the 5-epoch average to account for the periodic cost of adversarial updates.

| Method | Density | Noise | Trans. | Avg. | $T_1$ | $T_5$ |
|---|---|---|---|---|---|---|
| Vanilla | 70.35 | 55.87 | 75.32 | 67.18 | **47.52** | 45.82 |
| PointMixup | 66.32 | 76.62 | 77.54 | 73.49 | 92.52 | 91.27 |
| RSMix | **73.72** | 71.77 | 60.62 | 68.70 | 56.55 | 56.48 |
| PGD | 64.76 | 71.17 | 72.52 | 69.48 | 246.39 | 244.15 |
| PointCutMix-K | 69.08 | 71.97 | 71.22 | 70.75 | 78.46 | 76.51 |
| PointCutMix-R | 69.42 | **85.62** | 73.36 | 76.14 | 76.91 | 76.22 |
| AdaptPoint | 70.38 | 63.67 | 73.75 | 69.27 | 146.43 | 147.61 |
| PointPatchMix | 72.05 | 80.09 | 74.39 | 75.51 | 85.90 | 84.54 |
| **PSMix-S** | 70.79 | 78.73 | **83.61** | **77.71** | 88.73 | 114.62 |

## 4.3. Computational Complexity Analysis of PSMix

Table 4 reports the performance and runtime of PSMix-S compared to other data augmentation methods on ModelNet40-C using the Point-MAE backbone. To provide a fair metric for computational cost, we report the five-epoch average training time $T_5$, as our "Adversarial Rotation Optimization" is performed once every five epochs.

PSMix-S achieves the highest mean accuracy of $77.71\%$, outperforming the best spatial method, PointCutMix-R, by $1.57\%$ with a moderate increase in training time from $76.22s$ to $114.62s$ per epoch. Compared to generative methods, PSMix-S is considerably more efficient than PGD at $244.15s$ and AdaptPoint at $147.61s$. These results demonstrate that PSMix-S achieves a favorable balance between accuracy and computational cost. The advantage of PSMix-S is particularly pronounced on Transformation corruptions, where it obtains an accuracy of $83.61\%$, substantially outperforming PointMixup's $77.54\%$.

## 4.4. Ablation Study

We validate the effectiveness of three core components in PSMix-S on ModelNet40-C: **1) the Randomized Maximum SHT Degree** $L$, **2) the Hierarchical Coefficients Mixing**, and **3) the Adversarial Rotation Optimization**.

*Table 5.* Ablation study on ModelNet40-C using PointNeXt. We evaluate the contribution of randomized SHT degree $L$, hierarchical mixing, and adversarial rotation optimization.

| Setting | Density | Noise | Transformation | | Avg. |
|---|---|---|---|---|---|
| | | | Mean | ROT | |
| Vanilla | 64.28 | 60.14 | 64.12 | 37.52 | 62.85 |
| $L = 15$ | 67.72 | 82.55 | 82.66 | 68.84 | 77.64 |
| $L = 30$ | 67.76 | 82.86 | 84.51 | 72.53 | 78.38 |
| $L = 45$ | _68.59_ | _83.87_ | 83.36 | 69.65 | 78.61 |
| $L = 60$ | 68.00 | 83.32 | **84.73** | **74.35** | 78.68 |
| $L = 70$ | 67.38 | 82.85 | 83.97 | 71.80 | 78.07 |
| Linear Mix | **70.23** | 82.92 | 82.97 | 71.03 | _78.70_ |
| w/o Rotation | 66.89 | 82.09 | 81.14 | 64.67 | 76.70 |
| Spatial Rotation | 67.77 | 82.79 | 83.23 | 70.38 | 77.93 |
| Spectral Rotation | 68.03 | 83.62 | 84.37 | 72.73 | 78.68 |
| **PSMix-S** | 67.82 | **84.78** | _84.71_ | _73.50_ | **79.10** |

**Effect of Randomized Maximum SHT Degree** $L$. Rather than fixing the spectral resolution to a single value, PSMix-S dynamically samples $L \sim \mathcal{U}[15, 70]$ during training. To assess the effectiveness of this strategy, we evaluate model performance under fixed $L$ configurations, as reported in Table 5. Substituting dynamic sampling with any fixed value $L \in \{15, 30, 45, 60, 70\}$ consistently degrades average accuracy relative to the full PSMix-S. Mean accuracy increases

steadily as $L$ rises from 15 to 60 and reaches a peak of $78.68\%$. This behavior is expected because higher spectral resolutions capture more geometric detail, which benefits models trained on high-fidelity data. Further increases in $L$ lead to a slight decline, likely attributable to the model overfitting to redundant high-frequency information.

**Effect of Hierarchical Coefficients Mixing.** We compare our hierarchical strategy against a uniform "Linear Mix". As shown in Table 5, our approach outperforms the linear counterpart by $0.4\%$, demonstrating that respecting the distinct roles of spectral frequencies better maintains geometric integrity than uniform interpolation.

**Effect of Adversarial Rotation Optimization.** Table 5 compares PSMix-S against three baselines. First, the "w/o Rotation" variant suffers a $2.40\%$ drop in mean accuracy and a significant $8.83\%$ decline on ROT corruption, confirming the criticality of rotation-based augmentation. Second, we implement "Spatial Rotation" by applying rotation directly to the input point cloud. While this method yields reasonable performance, it still underperforms PSMix-S by $1.17\%$ in average accuracy and $3.12\%$ in ROT corruption, highlighting the superiority of spectral-domain manipulation. Finally, while random "Spectral Rotation" outperforms the spatial counterpart, it still trails PSMix-S by $0.42\%$ in average accuracy and $0.77\%$ in ROT corruption. These results indicate that our adversarial optimization goes beyond simple data augmentation. Instead, it actively generates challenging examples to drive the model toward learning more generalizable geometric features.

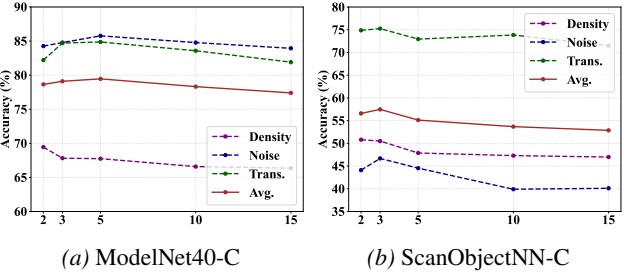

*(a)* ModelNet40-C     *(b)* ScanObjectNN-C

*Figure 4.* Sensitivity analysis of the mixing threshold $\tau$.

**Sensitivity Analysis of the Mixing Threshold** $\tau$. The impact of the transition parameter $\tau$ is evaluated across $\{2, 3, 5, 10, 15\}$, as illustrated in Figure 4. On ModelNet40-C, mean accuracy peaks at $79.46\%$ at $\tau = 5$ and declines gradually for larger values, indicating that well-structured shapes benefit from more low-frequency components establishing the base geometry before high-frequency details are introduced. On ScanObjectNN-C, the optimal result of $57.47\%$ is achieved at the smaller threshold $\tau = 3$, as noisy real-world scans are more sensitive to low-frequency overemphasis. Both trends confirm that a moderate $\tau$ best balances low- and high-frequency spectral mixing.

## 4.5. Extension to Large-Scale Scenes

To scale PSMix to large or dense point clouds, we introduce a patch-wise variant, **PSMix-P**, which applies spectral augmentation at the local patch level. Specifically, each input point cloud is partitioned into 8 non-overlapping patches, each containing 256 points. For every patch, we independently compute the local spectral coefficients $\hat{P}_i$ and $\hat{Q}_i$, where $i \in [1, 8]$. Following the PSMix-S, spectral rotation is applied to the coefficients of $\hat{Q}_i$ to produce $\hat{Q}'_i$. Hierarchical spectral mixing is then performed between corresponding coefficients $\hat{P}_i$ and $\hat{Q}'_i$, resulting in $\hat{P}_i^{mix}$, which integrates global spectral structure while preserving local geometric details. Finally, each mixed patch $\hat{P}_i^{mix}$ is reconstructed via the inverse SHT (ISHT), and points are sampled from the reconstructed patches to form the augmented point cloud $\tilde{P}$.

*Table 6.* Comparison of PSMix-P and PSMix-S performance on ScanObjectNN-C using the PointNeXt backbone.

| Method | Density | Noise | Transformation | Avg. |
|--------|---------|-------|----------------|------|
| PSMix-S | 50.50 | 46.69 | **75.25** | 57.47 |
| **PSMix-P** | **50.81** | **53.63** | 72.77 | **59.07** |

We conduct preliminary experiments on ScanObjectNN-C using the PointNeXt backbone to evaluate the effectiveness of PSMix-P. As reported in Table 6, PSMix-P yields improvements on Density ($+0.31\%$) and Noise ($+6.94\%$), while Transformation corruptions decrease slightly ($-2.48\%$), resulting in an average gain of $+1.6\%$. These results suggest that PSMix-P can be extended to large or dense point clouds, indicating its potential applicability beyond the standard PSMix-S framework.

## 5. Conclusion and Limitations

We presented PSMix, a spectral framework that mitigates geometric discontinuities in spatial mixup by performing rotation-aware hierarchical fusion on Spherical Harmonic coefficients, generating structurally coherent samples while preserving global topology. An adversarial rotation optimization strategy further improves orientation invariance via a minimax objective. Experiments on ModelNet40-C and ScanObjectNN-C demonstrate state-of-the-art robustness across multiple backbones, and PSMix-H serves as an orthogonal plug-in that yields consistent gains alongside existing spatial methods. We also introduce PSMix-P, a patch-wise variant whose preliminary results suggest potential for larger-scale point clouds. However, PSMix-P is currently validated only on object-level benchmarks, and the adversarial optimization introduces periodic training overhead. Future work will investigate more efficient training strategies and broader applications in 3D understanding.

## Acknowledgments

This work was supported in part by the National Natural Science Foundation of China under Grants 62506280, U22A2096, 62576261 and 62571395, in part by the China Postdoctoral Science Foundation under Grant Number 2025M771559, in part by the Postdoctoral Fellowship Program of CPSF under Grant Number GZB20250399, in part by Scientific and Technological Innovation Teams in Shaanxi Province under grant 2025RS-CXTD-011, in part by the Shaanxi Province Core Technology Research and Development Project under grant 2024QY2-GJHX-11, and in part by the CCF-Kuaishou Large Model Explorer Fund (NO. CCF-KuaiShou 2025006).

## Impact Statement

This paper presents work whose goal is to advance the field of machine learning. There are many potential societal consequences of our work, none of which we feel must be specifically highlighted here.

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

## A. Additional Experiments on the Point Cloud Mamba Backbone

To further evaluate the generality of PSMix, we compare it with other spatial-domain data augmentation methods, including the recent SinPoint (Bi et al., 2025), on the updated and more competitive Point Cloud Mamba (Zhang et al., 2025) backbone. The comparison results are summarized in Table 7.

*Table 7.* Classification accuracy (%) on ModelNet40-C with the Point Cloud Mamba Backbone.

| | Method | UNI | GAU | BG | IP | US | RBF | IR | DD | DI | SH | ROT | CO | FFD | OC | LD | Avg. |
|---|---|---|---|---|---|---|---|---|---|---|---|---|---|---|---|---|---|
| Point Cloud Mamba | Vanilla | 57.37 | 52.43 | 23.50 | 49.64 | 67.67 | 68.76 | 71.39 | 85.29 | 88.25 | 74.07 | 44.04 | 82.58 | 71.64 | 39.87 | 32.29 | 60.59 |
| | PointMixup | 79.13 | 76.94 | 38.29 | 73.26 | 82.50 | 82.66 | 83.02 | 86.22 | 87.07 | 79.90 | 49.92 | 86.22 | 77.43 | 34.12 | 29.09 | 69.72 |
| | RSMix | 33.10 | 36.30 | 86.75 | 83.47 | 50.97 | 56.04 | 58.23 | 90.11 | 90.03 | 66.41 | 30.06 | 88.90 | 59.97 | 43.56 | 42.46 | 61.09 |
| | PointCutMix-K | 31.00 | 30.51 | 86.83 | 78.48 | 44.53 | 69.17 | 72.16 | **90.64** | **90.32** | 73.06 | 36.18 | **90.64** | 70.18 | **47.12** | **48.06** | 63.92 |
| | PointCutMix-R | 73.34 | 73.18 | **90.52** | **89.51** | 76.09 | 73.66 | 75.04 | 90.03 | 89.47 | 73.70 | 44.17 | 87.56 | 73.26 | 42.02 | 43.48 | 73.00 |
| | SinPoint | 44.89 | 35.45 | 24.27 | 51.62 | 56.81 | 71.19 | 73.62 | 87.88 | 88.98 | 75.49 | 40.80 | 85.49 | 73.01 | 41.33 | 36.63 | 59.16 |
| | **PSMix-S(Ours)** | **81.85** | **81.24** | 48.99 | 75.53 | **83.51** | **85.98** | **87.40** | 87.07 | 88.37 | **83.63** | **66.37** | 83.67 | **84.00** | 41.17 | 40.48 | **74.61** |

The results show that PSMix-S achieves the highest mean accuracy of 74.61%, outperforming the best spatial-domain baseline, PointCutMix-R (73.00%), by 1.61%. In contrast, SinPoint under the same corruption robustness setting attains 59.16%, which is below the vanilla baseline (60.59%), suggesting that SinPoint primarily improves clean-data generalization rather than robustness. PSMix shows relatively consistent performance across corruption types, particularly for rotation and transformation perturbations, reflecting the intended effect of spectral hierarchical mixing in preserving global geometric structure while incorporating local details. These results confirm that PSMix effectively enhances robustness on modern backbones and functions as a versatile plug-in augmentation framework.

## B. More Visualization Results of PSMix

We present additional PSMix results on ModelNet40 (Wu et al., 2015) in Fig 5. The augmented instances simultaneously maintain the original sample's geometric features and integrate the global structure from a different sample.

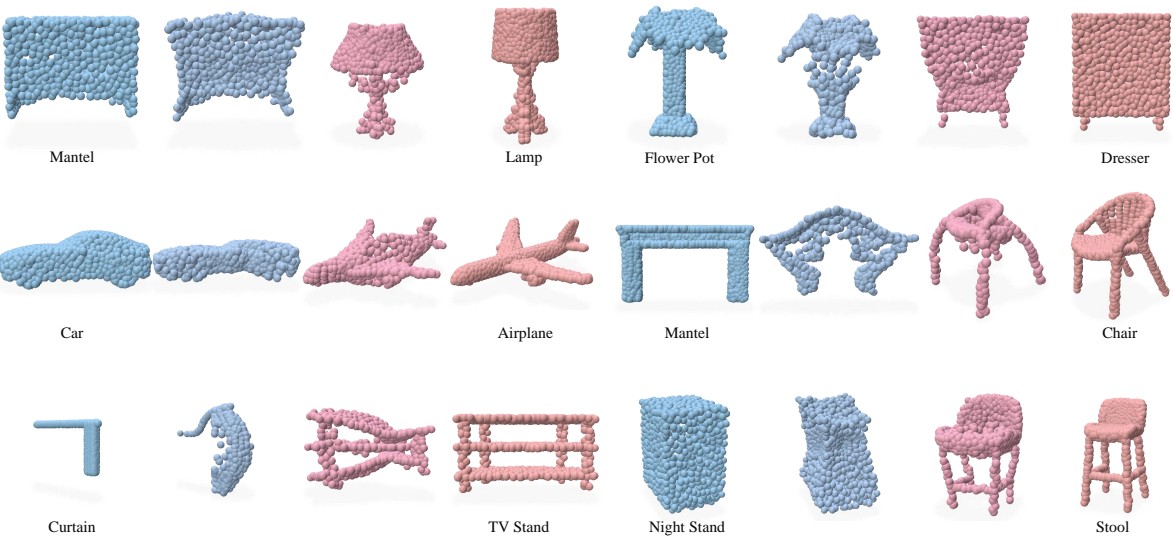

*Figure 5.* Additional visualization results from PSMix. The samples used for PSMix are all from the training set of ModelNet40 (Wu et al., 2015), each containing 1024 points.

## C. Comparable Methods in Experiments

In Sect. 4, we compare PSMix-H and PSMix-S with various 3D data augmentation methods on ModelNet40-C (Sun et al., 2021b) and ScanObjectNN-C datasets. The comparison includes mix-augmentation methods such as PointMixup (Chen et al., 2020), RSMix (Lee et al., 2021), and PointCutMix (Zhang et al., 2022), as well as generative augmentation approaches like PGD (Mirza et al., 2022) and Adaptpoint (Wang et al., 2023). Below, we provide a detailed overview of these methods along with their experimental configurations and hyperparameters.

**PointMixup.** PointMixup (Chen et al., 2020) enhances point cloud classification by introducing a data augmentation strategy based on optimal interpolation between unordered point sets. To address the lack of one-to-one correspondences in point clouds, PointMixup computes the shortest path assignment between two point clouds and interpolates both points and labels linearly. This ensures label consistency and enables the direct application of Mixup-style regularization to point cloud data. The method is theoretically proven to maintain consistent point assignments across interpolation ratios and is empirically validated to improve classification, few-shot learning, and semi-supervised tasks. The results reported are obtained using the publicly available implementation provided at `https://github.com/jiachens/ModelNet40-C`.

**RSMix.** Rigid Subset Mix (RSMix) (Lee et al., 2021) is a shape-preserving data augmentation method for 3D point clouds that mixes samples by copying and pasting Rigid Subsets (RS)—local clusters of neighbouring points defined by a spatial radius from a query point. This approach preserves structural integrity of the source data while enhancing diversity and regularization. Unlike interpolation-based methods (*e.g.*, PointMixup), RSMix avoids geometric distortion and maintains semantic consistency. RS scale is randomly varied to boost sample diversity, and the method is compatible with other augmentations. We reproduce results using the official implementation available at `https://github.com/jiachens/ModelNet40-C`.

**PGD.** Projected Gradient Descent (PGD) (Mirza et al., 2022) generates adversarial examples by iteratively applying gradient ascent to maximize the model's loss within a bounded perturbation region, typically under an $\ell_\infty$ or $\ell_2$ norm constraint. At each step, the input is updated in the direction of the gradient and projected back to the valid perturbation set. This method is widely regarded as a strong white-box attack for evaluating adversarial robustness. The critical attack hyperparameters include the step size $\alpha$, number of attack iterations $K$, and perturbation bound $\epsilon$, often set to $\alpha = 0.01$, $K = 20$, and $\epsilon = 0.05$ in 3D point cloud settings. The official implementation used to reproduce the results is available at `https://github.com/jiachens/ModelNet40-C`.

**PointCutMix.** PointCutMix (Zhang et al., 2022) enhances model performance and robustness by creating mixed point cloud samples through optimal assignment-based point replacement. Two variants are proposed: PointCutMix-R, which randomly replaces points in a point cloud with their optimal counterparts from another sample, and PointCutMix-K, which locally replaces the k nearest neighbours of a randomly chosen point. This approach improves object- and point-wise classification accuracy, especially for rare classes, and boosts robustness under point cloud attacks. The results reported are obtained using the publicly available implementation provided at `https://github.com/jiachens/ModelNet40-C`.

**AdaptPoint.** AdaptPoint (Wang et al., 2023) introduces a sample-adaptive auto-augmentation framework that enhances point cloud recognition robustness by simulating realistic corruptions. It employs an imitator network composed of a Deformation Controller and a Mask Controller to generate corrupted point clouds based on local geometry and structural information. The imitator is trained jointly with a discriminator, which ensures the generated samples follow the clean data distribution, and a classifier, which guides the difficulty of augmentations through error feedback. The framework requires no manual corruption design and progressively generates meaningful training data. The critical training-time hyperparameters for this approach include a learning rate of 0.001 and a batch size of 32. The official repository used to reproduce the results is available at `https://github.com/Roywangj/AdaptPoint`.

## D. Per Corruption Results of Joint Spatial-Spectral Data Augmentation Strategy

*Table 8.* Classification accuracy (%) is reported across all 15 corruptions in the ModelNet40-C dataset (Sun et al., 2021b). The results show the performance of backbone models trained on ModelNet40 (Wu et al., 2015) and tested on the corrupted dataset ModelNet40-C with a batch size of 32. "+PSMix-H" denotes a strategy where spatial domain data augmentation methods are integrated with our PSMix-H to achieve a joint spatial-spectral data augmentation. Green indicates improved accuracy, while red shows a decrease.

| | Method | Density | | | | | Noise | | | | | Transformation | | | | | Avg. |
|---|---|---|---|---|---|---|---|---|---|---|---|---|---|---|---|---|---|
| | | DD | DI | CO | OC | LD | UNI | GAU | BG | IP | US | RBF | IR | SH | ROT | FFD | |
| DGCNN | Vanilla | 72.48 | 83.02 | 75.68 | 33.06 | 11.70 | 79.45 | 73.41 | 61.66 | 61.83 | 67.09 | 77.75 | 80.38 | 84.64 | 59.15 | 80.10 | 66.76 |
| | PointMixup | 79.01 | 87.52 | 83.18 | 35.29 | 29.05 | 88.05 | 86.75 | 55.92 | 84.03 | 84.96 | 85.45 | 86.14 | 83.95 | 63.29 | 84.56 | 74.48 |
| | +PSMix-H | $81.32^{\uparrow2.31}$ | $87.80^{\uparrow0.28}$ | $83.51^{\uparrow0.33}$ | $39.47^{\uparrow4.18}$ | $35.21^{\uparrow6.16}$ | $90.19^{\uparrow2.14}$ | $89.38^{\uparrow2.63}$ | $78.40^{\uparrow22.48}$ | $88.33^{\uparrow4.30}$ | $87.40^{\uparrow2.44}$ | $89.22^{\uparrow3.77}$ | $89.18^{\uparrow3.04}$ | $87.64^{\uparrow3.69}$ | $70.10^{\uparrow6.81}$ | $86.99^{\uparrow2.43}$ | $78.94^{\uparrow4.46}$ |
| | RSMix | 88.01 | 89.30 | 89.18 | 40.68 | 34.68 | 84.23 | 79.78 | 87.31 | 88.04 | 78.24 | 80.51 | 81.68 | 86.46 | 59.32 | 83.63 | 76.74 |
| | +PSMix-H | $87.52^{\downarrow0.49}$ | $88.98^{\downarrow0.32}$ | $88.57^{\downarrow0.61}$ | $39.63^{\downarrow1.05}$ | $34.97^{\uparrow0.29}$ | $88.98^{\uparrow4.75}$ | $87.80^{\uparrow8.02}$ | $85.58^{\downarrow1.73}$ | $88.86^{\uparrow0.82}$ | $84.20^{\uparrow5.96}$ | $88.90^{\uparrow8.39}$ | $89.42^{\uparrow7.74}$ | $88.98^{\uparrow2.52}$ | $77.43^{\uparrow18.11}$ | $87.52^{\uparrow3.89}$ | $80.49^{\uparrow3.75}$ |
| | PointCutMix-K | 88.41 | 89.78 | 88.37 | 38.41 | 27.18 | 83.79 | 77.39 | 89.75 | 86.38 | 78.88 | 83.51 | 85.41 | 88.25 | 62.84 | 86.22 | 76.97 |
| | +PSMix-H | $88.05^{\downarrow0.36}$ | $89.99^{\uparrow0.21}$ | $89.91^{\uparrow1.54}$ | $40.11^{\uparrow1.70}$ | $36.26^{\uparrow9.08}$ | $90.19^{\uparrow6.40}$ | $88.49^{\uparrow11.10}$ | $89.42^{\downarrow0.33}$ | $88.98^{\uparrow2.60}$ | $85.41^{\uparrow6.53}$ | $90.11^{\uparrow6.60}$ | $90.15^{\uparrow4.74}$ | $90.07^{\uparrow1.82}$ | $80.39^{\uparrow17.55}$ | $89.42^{\uparrow3.20}$ | $81.80^{\uparrow4.83}$ |
| | PointCutMix-R | 87.44 | 88.98 | 86.99 | 36.99 | 29.41 | 85.74 | 83.75 | 89.71 | 88.61 | 81.07 | 83.87 | 85.05 | 85.57 | 62.19 | 84.11 | 77.30 |
| | +PSMix-H | $87.6^{\uparrow0.16}$ | $90.19^{\uparrow1.21}$ | $87.56^{\uparrow0.57}$ | $38.37^{\uparrow1.38}$ | $34.48^{\uparrow5.07}$ | $89.79^{\uparrow4.05}$ | $89.59^{\uparrow5.84}$ | $89.83^{\uparrow0.12}$ | $89.99^{\uparrow1.38}$ | $85.33^{\uparrow4.26}$ | $89.18^{\uparrow5.31}$ | $89.26^{\uparrow4.21}$ | $88.17^{\uparrow2.60}$ | $75.24^{\uparrow13.05}$ | $87.84^{\uparrow3.73}$ | $80.83^{\uparrow3.53}$ |
| PointNeXt | Vanilla | 81.40 | 81.89 | 82.33 | 42.87 | 32.90 | 59.81 | 50.20 | 53.93 | 63.37 | 73.38 | 68.64 | 70.54 | 74.84 | 37.52 | 69.08 | 62.85 |
| | Mixup | 86.75 | 87.56 | 87.12 | 42.71 | 34.20 | 86.22 | 84.52 | 63.86 | 84.24 | 88.21 | 85.17 | 86.63 | 82.66 | 54.13 | 83.63 | 75.84 |
| | +PSMix-H | $86.06^{\downarrow0.69}$ | $89.75^{\uparrow2.19}$ | $85.29^{\downarrow1.83}$ | $44.12^{\uparrow1.41}$ | $39.42^{\uparrow5.22}$ | $89.71^{\uparrow3.49}$ | $89.38^{\uparrow4.86}$ | $70.02^{\uparrow6.16}$ | $89.18^{\uparrow4.94}$ | $89.95^{\uparrow1.74}$ | $89.55^{\uparrow4.38}$ | $89.99^{\uparrow3.36}$ | $86.67^{\uparrow4.04}$ | $72.37^{\uparrow18.24}$ | $86.75^{\uparrow3.12}$ | $79.88^{\uparrow4.04}$ |
| | RSMix | 89.14 | 90.88 | 90.60 | 43.60 | 36.63 | 44.89 | 37.44 | 86.87 | 88.49 | 65.84 | 58.10 | 60.70 | 68.52 | 28.32 | 62.16 | 63.48 |
| | +PSMix-H | $89.75^{\uparrow0.61}$ | $90.56^{\downarrow0.32}$ | $89.79^{\downarrow0.81}$ | $47.93^{\uparrow4.33}$ | $46.52^{\uparrow9.89}$ | $87.03^{\uparrow42.14}$ | $85.82^{\uparrow48.38}$ | $86.59^{\downarrow0.28}$ | $89.06^{\uparrow0.57}$ | $87.32^{\uparrow21.48}$ | $87.64^{\uparrow29.54}$ | $88.98^{\uparrow28.28}$ | $84.52^{\uparrow16.00}$ | $59.68^{\uparrow31.36}$ | $85.58^{\uparrow23.42}$ | $80.45^{\uparrow16.97}$ |
| | PointCutMix-K | 86.02 | 87.28 | 88.05 | 46.31 | 36.55 | 62.40 | 53.24 | 89.79 | 79.13 | 81.07 | 73.58 | 75.41 | 75.97 | 38.45 | 75.00 | 69.88 |
| | +PSMix-H | $88.98^{\uparrow2.96}$ | $90.72^{\uparrow3.44}$ | $89.91^{\uparrow1.86}$ | $47.57^{\uparrow1.26}$ | $44.33^{\uparrow7.78}$ | $88.86^{\uparrow26.46}$ | $87.16^{\uparrow33.92}$ | $86.30^{\downarrow3.49}$ | $87.88^{\uparrow8.75}$ | $89.59^{\uparrow8.52}$ | $89.55^{\uparrow15.97}$ | $90.64^{\uparrow15.23}$ | $86.10^{\uparrow10.13}$ | $71.15^{\uparrow32.70}$ | $87.88^{\uparrow12.88}$ | $81.77^{\uparrow11.89}$ |
| | PointCutMix-R | 87.16 | 88.21 | 88.45 | 39.55 | 36.59 | 79.90 | 76.50 | 89.99 | 88.17 | 86.79 | 76.18 | 78.28 | 75.61 | 43.40 | 75.08 | 73.99 |
| | +PSMix-H | $87.72^{\uparrow0.56}$ | $89.55^{\uparrow1.34}$ | $87.16^{\downarrow1.29}$ | $41.37^{\uparrow1.82}$ | $38.74^{\uparrow2.15}$ | $89.67^{\uparrow9.77}$ | $89.26^{\uparrow12.76}$ | $87.03^{\downarrow2.96}$ | $90.68^{\uparrow2.51}$ | $88.98^{\uparrow2.19}$ | $89.42^{\uparrow13.24}$ | $89.91^{\uparrow11.63}$ | $85.94^{\uparrow10.33}$ | $71.52^{\uparrow28.12}$ | $87.20^{\uparrow12.12}$ | $80.94^{\uparrow6.95}$ |

In Sect. 4.3, we present the results to evaluate the effectiveness of the joint spatial-spectral data augmentation strategy. Here, we provide a detailed breakdown of the performance across individual corruptions to analyze the contribution of our PSMix-H. This analysis highlights how PSMix-H improves robustness across various corruption types.

Table 8 details the performance comparison on the ModelNet40-C (Sun et al., 2021b) across 15 corruptions. The results show that integrating PSMix-H with spatial augmentation methods generally enhances classification accuracy over standalone methods for both DGCNN (Wang et al., 2019) and PointNeXt (Qian et al., 2022) backbones. With the DGCNN backbone, PointMixup (Chen et al., 2020) alone achieves a modest mean accuracy of 74.48% over the 66.76% baseline, despite a significant performance drop under BG corruption to 55.92% versus the baseline's 61.66%. Introducing PSMix-H further raises the mean accuracy by 4.46%, demonstrating effective gains from spectral mixing, with notable improvements under BG corruption, reaching 78.40% compared to the PointMixup's 55.92%. For PointCutMix-K (Zhang et al., 2022), the addition of PSMix-H improves mean accuracy from 76.97% to 81.80% and increases accuracy on ROT from 62.84% to 80.39%. On the PointNeXt backbone, the integration of PSMix-H also frequently leads to accuracy gains. For example, with RSMix (Lee et al., 2021), integrating PSMix-H increased accuracy on UNI from 44.89% to 87.03% and on ROT from 28.32% to 59.68%. When combined with PointCutMix-K (Zhang et al., 2022), PSMix-H improves performance on GAU from 53.24% to 87.16%.

While occasional minor performance decreases occur for specific method-corruption combinations on DGCNN and PointNeXt, the overall trend remains positive. For both backbones, incorporating PSMix-H consistently yields improvements across all transformation corruptions, including RBF, IR, SH, ROT, and FFD. Furthermore, the combination of PointCutMix-K with PSMix-H achieves the highest average accuracy on DGCNN at 81.80% and on PointNeXt at 81.77%.

# E. Per Corruption Results of Ablation Study

*Table 9.* Ablation study conducted on all corruptions of ModelNet40-C involves dissecting the core strategies of PSMix-S. This includes randomized maximum SHT degree $L$, hierarchical coefficients mixing strategy and adversarial rotation optimization strategy.

| Setting | Density | | | | | Noise | | | | | Transformation | | | | | Avg. |
|---|---|---|---|---|---|---|---|---|---|---|---|---|---|---|---|---|
| | DD | DI | CO | OC | LD | UNI | GAU | BG | IP | US | RBF | IR | SH | ROT | FFD | |
| Vanilla | 81.40 | 81.89 | 82.33 | 42.87 | 32.90 | 59.81 | 50.20 | 53.93 | 63.37 | 73.38 | 68.64 | 70.54 | 74.84 | 37.52 | 69.08 | 62.85 |
| Fixed $L = 30$ | 85.70 | 89.38 | 84.89 | 42.26 | 36.55 | 87.84 | 86.47 | 67.02 | 83.67 | **89.30** | **88.65** | 89.47 | 85.29 | 72.53 | 86.63 | 78.38 |
| Fixed $L = 45$ | 85.90 | 89.22 | 85.66 | 43.72 | 38.45 | 87.28 | 85.90 | 74.07 | 83.10 | 88.98 | 88.25 | 88.74 | 83.95 | 69.65 | 86.22 | 78.61 |
| Fixed $L = 60$ | 86.14 | 87.88 | 86.30 | 43.72 | 35.94 | 87.20 | 85.33 | 72.73 | 82.62 | 88.70 | 87.97 | 89.14 | 85.37 | **74.35** | 86.83 | 78.68 |
| Linear Mix | **87.52** | 86.75 | **87.56** | 47.93 | **41.37** | 88.29 | 86.71 | 69.49 | 81.60 | 88.49 | 86.02 | 87.84 | 85.01 | 71.03 | 84.97 | 78.70 |
| No Rotation | 85.21 | 88.82 | 83.18 | 42.75 | 34.48 | 86.79 | 84.93 | 68.31 | 81.89 | 88.53 | 86.71 | 86.67 | 83.47 | 64.67 | 84.16 | 76.70 |
| Spatial Rotation | 85.90 | 89.22 | 84.20 | 43.88 | 35.66 | 87.20 | 85.05 | 72.29 | 81.32 | 88.09 | 87.20 | 88.78 | 83.02 | 70.38 | 86.79 | 77.93 |
| Spectral Rotation | 85.45 | 89.10 | 85.49 | 42.79 | 37.32 | 88.29 | 86.63 | 69.69 | **84.81** | 88.70 | 87.80 | 88.90 | **86.26** | 72.73 | 86.18 | 78.68 |
| PSMix-S | 85.98 | **89.71** | 84.40 | 42.71 | 36.30 | **88.45** | 86.71 | **75.57** | 84.16 | 89.02 | 87.80 | **89.55** | 85.37 | 73.50 | **87.32** | **79.10** |

In Sect. 4.4, we present the results of ablation studies to evaluate the effectiveness of the components of PSMix-S. Here, we detail the performance across individual corruptions to analyze the contribution of each component. This analysis highlights how the randomized maximum SHT degree $L$ enhances multi-resolution spectral representations learning, the hierarchical coefficients mixing strategy preserves geometric integrity, and the adversarial rotation optimization strategy improve robustness across various corruption types.

Table 9 presents the performance analysis of PSMix-S variants on ModelNet40-C (Sun et al., 2021b) across 15 corruption types. The complete PSMix-S configuration, incorporating all components, achieves the highest mean accuracy of 79.10%, demonstrating notable strength particularly under GAU noise (86.71%) and ROT corruption (73.50%). When the randomized maximum SHT degree $L$ is replaced with fixed $L$ values of $30, 45$ and $60$, the average accuracy decreases by 0.72%, 0.49%, and 0.42%, respectively. This underscores the benefit of leveraging multi-resolution spectral information for robust feature learning. Replacing the hierarchical coefficients mixing strategy with "Linear Mix", which entails uniform spectral coefficient interpolation, leads to a 0.4% reduction in average accuracy. This illustrates that hierarchical fusion more effectively upholds geometric integrity by acknowledging the distinct contributions of various spectral frequencies. However, we observe that the Linear Mix outperforms the full PSMix-S by 5.22% on OC and 5.07% on LD. This occurs because simple linear spectral mixing superimposes the point clouds spatially, partially simulating occlusion and sparse density, which improves performance against density corruptions. Finally, substituting the adversarial rotation optimization strategy with "No Rotation", "Spatial Rotation", and "Spectral Rotation" resulted in respective decreases of 2.40%, 1.17%, and 0.42% in average classification accuracy, and 8.83%, 3.12%, and 0.77% under ROT corruption. These results indicate that adversarial optimization is not merely performing various spectral rotations. Instead, it actively unearths more challenging and informative samples, thereby driving the model to develop more robust and generalizable features.

