# OpenReview forum: "PSMix: Robust Point Cloud Recognition through  Spectral Domain Mixing"
_ICML.cc/2026/Conference — ICML 2026 regular_

### Official Review · Reviewer_BsMt · 2026-03-02

**Soundness:** 2
**Presentation:** 3
**Significance:** 2
**Originality:** 2
**Overall Recommendation:** 4
**Confidence:** 5

**Summary:**

This paper proposes a mixup-style data augmentation method for point cloud recognition. The method performs sample mixing in the spectral domain rather than the spatial domain, which is intended to reduce local artifacts caused by unnatural spatial partitioning. Experimental results show that it achieves higher accuracy compared to previous augmentation methods.

**Compliance With Llm Reviewing Policy:**

Affirmed.

**Final Justification:**

I thank the authors for their detailed and constructive rebuttal. My main concerns have been largely addressed. I encourage the authors to explicitly discuss when and why spectral mixing is advantageous, and when spatial methods may still be preferable. Overall, I lean toward a Weak Accept.

**Key Questions For Authors:**

Please provide a more thorough comparison and analysis with prior spatial-domain augmentation methods, including PointPatchMix and SinPoint, and clearly articulate the advantages of spectral-domain enhancement over spatial-domain strategies.

**Limitations:**

yes.

**Strengths And Weaknesses:**

## Strengths

1. The paper performs sample mixing in the spectral domain rather than the spatial domain, aiming to reduce the disruption of intrinsic topological consistency caused by spatial interpolation.

2. The use of the Spherical Harmonic Transform (SHT) instead of the Graph Fourier Transform (GFT) provides a universal spectral basis, enabling mathematically valid cross-sample mixing.

3. The experimental results show certain improvements over existing spatial augmentation methods on corruption benchmarks.


* * *

## Weaknesses

1. Avoiding disruption of the intrinsic topological consistency is presented as the core motivation of the paper. However, based on the visualizations shown in Figures 1, 2, and 5, the generated samples still appear to exhibit irregular point distributions and local structural artifacts. In addition, the synthesized shapes lack clear semantic interpretability. A more detailed qualitative analysis would help substantiate the claimed geometric coherence.

2. There have been multiple prior works in the spatial domain that aim to preserve intrinsic topological consistency. For example, PointPatchMix [1] preserves local semantic structures through patch-level mixing to mitigate the adverse effects of spatial interpolation on feature extraction. However, the paper provides only a limited comparison with PointPatchMix in Table 4, and the performance gains are not substantial. In addition, PSMix incorporates additional rotation augmentation, while PointPatchMix does not include such augmentation. Therefore, under Transformation corruptions, the reported improvement may partially stem from this additional augmentation rather than from the spectral mixing strategy itself.

3. In addition, SinPoint [2] proposes a continuous deformation strategy in the spatial domain based on a homeomorphic mapping to preserve topological consistency. Compared with PointPatchMix, SinPoint offers a more principled treatment of structural continuity via homeomorphic mapping. However, SinPoint is not discussed in the paper. Including a comparison or discussion of such spatial-domain approaches would provide a more complete positioning of the proposed method.

4. Although the authors demonstrate that PSMix can be combined with spatial domain augmentation methods, the computational cost of the spectral transformation and adversarial rotation optimization is relatively high. This makes it difficult to clearly justify whether replacing spatial domain mixing with spectral domain mixing is the most practical approach for improving intrinsic topological consistency.


* * *

## Minor Weaknesses

5. The experimental section mainly focuses on numerical comparisons. A deeper analysis explaining why spectral mixing improves robustness under specific corruption types (e.g., Density, Noise, Transformation) would improve the clarity of the contribution.

6. In Table 4, the reported $T_5$ suggests that PSMix requires substantially more training time than some spatial augmentation methods (e.g., PointPatchMix [1]). Does this imply that the total training time increases by approximately 35%? Furthermore, when combining PSMix with spatial methods (as in Table 2), is the runtime approximately additive? Providing corresponding training time comparisons for the joint spatial-spectral setting would make the evaluation more complete.


* * *

## References

[1] Wang, Yi, et al. "Pointpatchmix: Point cloud mixing with patch scoring." Proceedings of the AAAI Conference on Artificial Intelligence. Vol. 38. No. 6. 2024.

[2] Bi, Jian, et al. "Rethinking Point Cloud Data Augmentation: Topologically Consistent Deformation." Forty-second International Conference on Machine Learning. 2025.

---

> ### Author Rebuttal · Authors · 2026-03-31
>
> We sincerely thank the reviewer for their thoughtful and constructive feedback. Below, we address each concern in detail:
>
> ---
>
> **W1: Geometric Coherence of Augmented Samples.**
>
> We clarify that our claim of geometric coherence is relative to prior mixup-based methods, not absolute. As shown in Fig. 1, compared with rsmix, cutmix, PSMix significantly reduces the sharp discontinuities, fragmented regions, and cut-and-paste artifacts, yielding smoother mixed samples overall. Some local irregularity may still remain, as PSMix must combine two point clouds with potentially different semantics — making perfect geometric continuity harder than in single-shape methods like SinPoint. Our goal is thus better geometric consistency within the mixup setting, not perfectly interpretable shapes, and we will clarify this further in the revised paper.
>
> ---
>
> **W2: Comparison with PointPatchMix on Point-MAE.**
>
> PointPatchMix is tightly coupled with Point-MAE, relying on a pretrained transformer for patch scoring. Since **the official scoring module is unavailable**, our comparison is based on our own reproduction and conducted on Point-MAE only, consistent with its original design.
>
> For the concern that the gains may mainly come from the added rotation augmentation, our ablation suggests otherwise. On ModelNet40-C with PointNeXt, Spatial Rotation achieves **$77.93\%$**, Spectral Rotation achieves **$78.68\%$**, and the full **PSMix-S** further improves to **$79.10\%$**, indicating that both rotation-aware design and spectral hierarchical mixing contribute. We further conduct a rotation ablation on Point-MAE:
>
> |Method|Avg.|$T_1$|$T_5$|
> |:---|:---:|:---:|:---:|
> |PointPatchMix|75.51|85.90|84.54|
> |PSMix-S w/o rot|77.15|**61.82**|**58.38**|
> |PSMix-S|77.71|88.73|114.62|
> |PointPatchMix+PSMix-H|**78.40**|238.02|265.4|
>
> Even without rotation, PSMix-S w/o rot already outperforms PointPatchMix (**$77.15\%$** vs. **$75.51\%$**).  Adding rotation further improves to **$77.71\%$**. This indicates that the gain does **not** mainly come from rotation augmentation; rather, the **spectral mixing strategy** is the primary source of improvement, while rotation provides an additional boost.
>
> ---
>
> **W3: Comparison with SinPoint on Point-Mamba.**
>
> Due to space constraints, please refer to our response to **Reviewer Y442 (W1)** for a detailed discussion.
>
> ---
>
> **W4: Computational Cost and Advantages of Spectral-domain Mixing**
>
> Due to space constraints, please refer to our response to **Reviewer 1NpB (L1)** for a detailed discussion on computational cost.
>
> Our additional Point-MAE ablation (W2) shows **PSMix-S w/o rot** already outperforms **PointPatchMix** in **Avg.** (**$77.15\%$ vs. $75.51\%$**) while also reducing runtime (**$T_1$: $61.82$ vs. $85.90$; $T_5$: $58.38$ vs. $84.54$**). This suggests that **spectral mixing is a practical alternative to spatial mixing** for improving topological consistency, while adversarial rotation serves as an additional enhancement when stronger robustness is needed.
>
> ---
>
> **W5: Analysis of Robustness under Different Corruption Types.**
>
> In short, PSMix improves robustness for different reasons across corruption types. For **Transformation** corruptions, the gain mainly comes from **rotation-aware spectral mixing**, which preserves global structure while improving invariance to orientation and deformation. For **Noise** corruptions, spectral mixing is less sensitive to local perturbations than direct spatial replacement. For **Density** corruptions, PSMix helps preserve the main geometry under sparsification, although its advantage is less dominant in corruption types that are closer to explicit occlusion or point removal. We will revise the paper to explain these category-specific effects more clearly.
>
> ---
>
> **W6: Training Time Overhead in Spatial-Spectral Joint Augmentation.**
>
> Since PSMix is used only as a **training-time augmentation**, its extra cost directly increases the total training time. For the joint spatial-spectral setting in W2’s Table, the runtime is also substantially higher, but **not simply additive**. This is because the combined method executes **both augmentations sequentially**, without shared intermediate computation. In particular, PointPatchMix introduces extra cost from **patch partition and transformer-based patch scoring**, while PSMix-H further requires **spectral transform, hierarchical mixing, and reconstruction**, leading to a much larger overall training overhead. Importantly, this cost appears **only during training**; **inference requires no extra computation** and remains the same as the vanilla backbone. Meanwhile, the joint setting improves the average accuracy from **75.51\%** to **78.40\%**, i.e., by **2.89 \%** over PointPatchMix. We will clarify this training-cost/robustness trade-off more explicitly in the revised paper.

---

> > ### Author Rebuttal · Reviewer_BsMt · 2026-04-02
> >
> > We thank the authors for their detailed rebuttal. However, after reading it, I still find that the manuscript lacks sufficient mechanistic evidence to justify why spatial partition is suboptimal and why geometric consistency in the spectral domain is fundamentally better.
> >
> > In addition, the rigor of the experimental setup and the fairness of comparisons remain limited. The responses to W2 and W5 do not provide convincing evidence or analysis to demonstrate the actual significance of geometric consistency compared to existing methods.
> >
> > Specifically, when comparing the full PSMix model with PointPatchMix, PSMix shows clear improvements under Transformation corruptions, even though it performs worse under Density and Noise. However, when rotation augmentation is removed, the average performance remains nearly unchanged. This makes it unclear whether the gains come from spectral mixing, rotation augmentation, or their interaction.
> >
> > Therefore, both the potential synergy and possible interference between spectral mixing and rotation augmentation are not well understood.
> >
> > These observations suggest that the current design may suffer from instability across datasets, corruption types, and model architectures, which weakens the claimed benefits of preserving geometric consistency.
> >
> > While the paper presents a meaningful attempt to explore spectral-domain augmentation for point cloud recognition and demonstrates certain empirical improvements, I believe the manuscript still requires major revisions in terms of research gap analysis, motivation justification, and fair and comprehensive experimental comparisons, and is not suitable for camera-ready at this stage.
> >
> > ### ACK Edit1
> >
> > ---
> >
> > We thank the authors for their detailed rebuttal and the additional per-corruption results. The clarifications and expanded table are helpful.
> >
> > First, the new results further confirm a noticeable bias toward Transformation corruptions. Compared with PointPatchMix, PSMix-S w/o rotation already derives most of its average gain from the Transformation group, while its performance on Noise and Density is lower. After adding rotation, the additional improvements are again mainly concentrated on transformation-sensitive cases (e.g., ROT and SH), with limited or negative changes on other categories. This suggests that the overall gain is not uniformly distributed. I recommend that the authors explicitly acknowledge and discuss this bias in the paper, rather than implying a balanced robustness improvement, as in (2).
> >
> > Second, the results also suggest that spatial-partition-based methods are not universally harmful. In particular, under BG corruption, most spatial methods remain significantly stronger, while PSMix shows a clear drop. This indicates that the artifacts introduced by spatial partitioning may be beneficial in certain scenarios, rather than universally detrimental as suggested in the current manuscript.
> >
> > Overall, I believe the paper would benefit from a more balanced discussion of the strengths and limitations of both approaches. In particular, the authors should more clearly articulate when and why spectral mixing is advantageous, and when spatial methods may still be preferable.
> >
> > ### ACK Edit2
> >
> > ---

---

> > > ### Author Response · Authors · 2026-04-02
> > >
> > > We thank the reviewer for the follow-up comment. Due to the strict space limit, both the original Table 4 in the paper and compressed summary in W2 may have been misleading. We apologize for this lack of clarity.
> > >
> > > ---
> > >
> > > (1) The gain is not only from Transformation corruptions.
> > >
> > > When the full per-corruption results are expanded, the improvement of PSMix over PointPatchMix is **not** limited to Transformation. The complete Point-MAE results are:
> > >
> > > ||UNI|GAU|BG|IP|US|RBF|IR|DD|DI|SH|ROT|CO|FFD|OC|LD|Avg|
> > > |-|--:|--:|--:|--:|--:|--:|--:|--:|--:|--:|--:|--:|--:|--:|--:|--:|
> > > |PointPatchMix|77.03|74.84|**85.05**|82.37*|81.16|81.28|81.81|**89.63**|**90.19**|80.06|49.76|**91.13**|79.05|**46.64**|42.67*|75.51|
> > > |PSMix-S w/o rot|86.30*|84.35*|59.40*|81.19|**83.87**|**87.88**|**88.89**|88.93|89.91|84.31*|62.43*|88.85*|**85.57**|44.32*|41.12|77.15*|
> > > |PSMix-S|**86.39**|**85.62**|56.65|**82.41**|82.58*|87.80*|88.78*|89.14*|90.15*|**86.06**|**69.89**|88.01|85.53*|43.40|**43.27**|**77.71**|
> > >
> > > These full results show that the average gain does **not** come only from Transformation. Compared with PointPatchMix, **PSMix-S w/o rot is already better on 8/15 corruptions**, and the full **PSMix-S is better on 10/15 corruptions**. PointPatchMix is mainly particularly strong on **BG**, where it exceeds **PSMix-S w/o rot by 25.65** and **PSMix-S by 28.40**, and also remains better on a few corruption types such as **CO/OC** and slightly on some density-related cases. In contrast, PSMix improves a broader set of corruptions, including **UNI, GAU, US, RBF, IR, SH, ROT, FFD**, and additionally improves **IP** and **LD** in the full model. Therefore, the final mean improvement comes from **multiple corruption types**, rather than from Transformation alone.
> > >
> > > In other words, PointPatchMix has a **more localized advantage** (especially on background noise), while PSMix provides a **broader overall gain profile**, which is exactly why its average accuracy is higher.
> > >
> > > ---
> > >
> > > (2) Removing rotation but still performing well indicates that the main gain comes from spectral mixing.
> > >
> > > We agree that this point should be interpreted carefully. In fact, the observation that **PSMix-S w/o rot already outperforms PointPatchMix (77.15 vs. 75.51)** directly supports that the **primary gain comes from spectral mixing**, not from adversarial rotation. If rotation were the dominant factor, removing it should have largely eliminated the gain, which is not what we observe.
> > >
> > > The role of rotation is therefore better understood as an **additional enhancement**, mainly strengthening robustness on geometry-sensitive corruptions, especially **rotation-related / deformation-related** cases. This is also consistent with the per-corruption table above: adding rotation further improves several transformation-sensitive corruptions, most notably **ROT** (**62.43 → 69.89**) and also **SH** (**84.31 → 86.06**), while the average gain from w/o rot to full remains modest (**77.15 → 77.71**). Thus, the evidence suggests a clear decomposition:
> > >
> > > - **spectral mixing** provides the main average-performance improvement;
> > > - **rotation augmentation** provides an additional boost, concentrated mainly on transformation-sensitive corruptions.
> > >
> > > So the results are not unclear; rather, they indicate that the two components play **different roles**, with spectral mixing being the principal source of improvement and rotation serving as an auxiliary robustness enhancer.
> > >
> > > ---
> > >
> > > (3) The method does not show instability across datasets, corruption types, and architectures.
> > >
> > > We respectfully disagree with the claim of instability. Based on the reported results, PSMix is consistently strong across **2 datasets** (**ModelNet40-C** and **ScanObjectNN-C**) and **4 backbones**, covering traditional backbones, transformer-based backbones, and mamba-based backbones. Across these settings, PSMix remains consistently competitive, and achieves **state-of-the-art** across **multiple backbones** rather than only on a single architecture.
> > >
> > > The margin naturally varies across corruption types and architectures, but such variation is expected for any augmentation method and does not imply instability. More importantly, the **overall performance remains consistently favorable** across both datasets and across traditional / transformer-based / mamba-based backbones. We therefore believe the evidence supports that the proposed design is **robust and broadly effective**, rather than unstable.
> > >
> > > ---
> > >
> > > ### Response Update
> > >
> > > Thank you for the thoughtful follow-up and for the constructive suggestions. We agree that a more balanced discussion is important, and we will revise the paper to explicitly clarify the transformation-oriented strength of PSMix, as well as the scenarios where spatial methods may still be preferable.

---

### Official Review · Reviewer_1NpB · 2026-03-05

**Soundness:** 3
**Presentation:** 2
**Significance:** 3
**Originality:** 2
**Overall Recommendation:** 4
**Confidence:** 3

**Summary:**

The paper addresses the data augmentation problem in point cloud recognition, pointing out that traditional spatial domain Mixup strategies often destroy geometric integrity and produce physically unrealistic samples.To address this limitation, this paper proposes a novel framework called PSMix, which shifts the hybrid paradigm from the spatial domain to the spectral domain based on Spherical Harmonic Transform(SHT).PSMix performs rotation-aware hierarchical blending of spectral coefficients, thereby increasing local diversity while preserving the global topology of the object.The paper also introduces an adversarial rotation optimization strategy to further improve the model's generalization ability by mining more challenging samples.This paper presents experiments on ModelNet40, ScanObjectNN, and the ModelNet40-C dataset specifically designed for robustness evaluation. Results show that PSMix achieves performance improvements across various mainstream backbone networks, particularly excelling in handling complex disturbances such as noise, occlusion, and rotation. Ablation experiments validate the crucial roles of hierarchical spectral mixing and adversarial rotation strategies in maintaining geometric consistency.The paper is clearly structured and logically rigorous, providing a solid theoretical foundation for the proposed method through mathematical formulas. The figures and tables in the paper are also quite intuitive.

**Compliance With Llm Reviewing Policy:**

Affirmed.

**Key Questions For Authors:**

In some cases, the effect is significant, while in others the effect is not ideal. What do you think is the reason?

**Limitations:**

The proposed method has a significantly higher computational cost, which may not be suitable for resource-constrained situations. Furthermore, the author points out that the global spherical harmonic transformation remains non-trivial even in large-scale scenarios, which is also a limitation.

**Strengths And Weaknesses:**

## Strength：

This paper notes that traditional data augmentation strategies can compromise geometric integrity, thus shifting the spatial domain mixup to the spectral domain based on spherical harmonic transform. Furthermore, it introduces an adversarial rotation optimization strategy, demonstrating considerable innovation.Furthermore, it is supported by rigorous mathematics and provides a detailed explanation of Spherical Harmonics Transform.The experimental section of the article is quite comprehensive, supplemented by ablation experiments, which strongly demonstrates the effectiveness of the proposed method.

## Weakness：

This paper lacks some originality: SHT is an existing method, and the concepts of low frequency representing contour and high frequency representing detail have also been discovered. This paper, while meaningful in introducing SHT into this direction, feels somewhat lacking in innovation.Another drawback is that the experimental results are slightly inferior in some situations: for example, the BG column in Table 1 and the GAU column in Table 2. These situations require consideration of experimental limitations.

---

> ### Author Rebuttal · Authors · 2026-03-31
>
> We sincerely thank the reviewer for their thoughtful and constructive feedback. Below, we address each concern in detail:
>
> ---
>
> **W1: Novelty Relative to Prior Work.**
>
> Prior spectral methods (HSP [1], GSDA [2], Texture [3]) rely on sample-specific spectral bases (e.g., GFT, Mesh Laplace). Since different point clouds yield different graphs, their eigenbases are **not aligned**, making cross-sample coefficient mixing mathematically ill-posed. Our core contribution is adopting SHT as a **universal, input-independent** basis, enabling principled cross-sample spectral mixing for the first time. Crucially, this is not merely replacing GFT with SHT; we design a full pipeline around the unique properties of SHT: (1) **Wigner D-matrix rotation** exploiting the rotation-equivariance of SHT for differentiable, gimbal-lock-free spectral rotation; (2) **hierarchical frequency-adaptive mixing** leveraging the multi-resolution structure of spherical harmonics; and (3) **adversarial rotation optimization** that actively generates challenging orientations via a minimax objective. These components are each enabled by the specific mathematical properties of SHT and together constitute a novel augmentation framework that goes well beyond prior spectral analyses of point clouds. We will further clarify these aspects in the revised paper.
>
> ---
>
> **W2&Q: Corruption-Specific Performance Variation and Limitations.**
>
> We agree that PSMix is **not uniformly optimal across all corruption types**, and we will clarify this limitation in the revised paper. Our method is designed around **rotation-aware hierarchical mixing in the spectral domain**, which mainly preserves **global structural topology** and is therefore more naturally suited to corruptions that change geometry or orientation. Consistent with this design, PSMix shows particularly strong gains on transformation corruptions such as **RBF, IR, FFD, and ROT**, while being less dominant on some noise corruptions.
>
> For instance, **BG** corruption introduces background noise rather than geometric deformation. Since PSMix focuses on structurally coherent spectral mixing instead of explicitly simulating noise, it is less specialized for this type. Nevertheless, PSMix-S still improves substantially over the vanilla baseline on ModelNet40-C, raising BG accuracy from **$61.66\%$ to $72.20\%$** on DGCNN and from **$53.93\%$ to $75.57\%$** on PointNeXt. We will make this corruption-specific analysis more explicit in the revised paper.
>
> ---
>
> **L1: Computational Cost of the Proposed Method.**
>
> PSMix is a **training-time data augmentation method**, so its additional cost is incurred **only during training**, with **zero extra inference latency** after deployment. As reported in Table 4, PSMix-S increases training time from **$76.22$ s/epoch** (PointCutMix-R) to **$114.62$ s/epoch**, while remaining more efficient than generative methods such as AdaptPoint (**$147.61$ s/epoch**) and PGD (**$244.15$ s/epoch**). We agree that this training overhead may be less suitable for resource-constrained settings, and we will clarify this trade-off in the revised paper.
>
> ---
>
> **L2: Potential Extension to Large-Scale Scenes.**
>
> We agree that applying global SHT to large-scale, unbounded scenes is non-trivial, likely requiring **block-wise spectral processing**. To explore this, we implemented **PSMix-P**, which partitions each point cloud into **8 local patches** (256 points each), applies SHT per patch, performs spectral rotation and hierarchical mixing on local coefficients, and reconstructs via ISHT. Results on **ScanObjectNN-C** with **PointNeXt**:
>
> |Setting|Density|Noise|Trans|Mean|
> |---|---:|---:|---:|---:|
> |PSMix-S|50.50|46.69|**75.25**|57.47|
> |PSMix-P|**50.81**|**53.63**|72.77|**59.07**|
>
> PSMix-P improves Noise ($+6.94\%$) substantially, increasing overall mean from $57.47\%$ to $59.07\%$. Meanwhile, Transformation slightly drops, suggesting that patch-wise spectral processing strengthens robustness to local perturbations but weakens the benefit of global spectral modeling. These results provide promising initial evidence for block-wise spectral extensions.
>
> ---
>
> [1] *Hypergraph Spectral Analysis and Processing in 3D Point Cloud.* TIP 2021.
>
> [2] *Exploring the Devil in Graph Spectral Domain for 3D Point Cloud Attacks.* ECCV 2020.
>
> [3] *Texture: Text-guided Texturing of 3D Shapes.* ACM SIGGRAPH 2023.

---

> > ### Author Rebuttal · Reviewer_1NpB · 2026-04-03
> >
> > Thank you to the authors for their detailed rebuttal and supplementary experiments. Your explanation of how SHT, as a universal basis, solves the ill-posed problem across sample mixtures clarifies the theoretical innovation more clearly than in the initial draft. I also appreciate your clarification that computational cost is limited to the training phase, and your analysis of the performance under non-geometric deformation interferences such as BG. Furthermore, you implemented a block-based processing version (PSMix-P) in a very short time and verified its potential for scaling to large-scale scenes on ScanObjectNN-C. Your responses have well addressed my main concerns; please be sure to include these discussions on applicability boundaries and the preliminary PSMix-P experimental results in the final revised version.

---

> > > ### Author Response · Authors · 2026-04-03
> > >
> > > Thank you for the positive feedback and helpful suggestion. We will include the discussion on applicability boundaries and the preliminary PSMix-P results in the final revised version.

---

### Official Review · Reviewer_Y442 · 2026-03-11

**Soundness:** 3
**Presentation:** 3
**Significance:** 3
**Originality:** 3
**Overall Recommendation:** 4
**Confidence:** 3

**Summary:**

This paper proposes a spectral-domain data augmentation framework named PSMix to improve the robustness of point cloud recognition.The method converts point clouds into the spectral domain via spherical harmonic transform and performs rotation-aware hierarchical mixing on the spectral coefficients, avoiding the geometric discontinuities caused by spatial-domain mixing methods. This paper also introduces an adversarial rotation optimization strategy, which dynamically identifies the most challenging rotation parameters through a minimax objective, enabling the model to learn features robust to severe geometric transformations. Experiments are conducted on two benchmarks, ModelNet-C and ScanObjectNN-C, which validate the effectiveness of the proposed method.

**Compliance With Llm Reviewing Policy:**

Affirmed.

**Key Questions For Authors:**

please see weaknesses.

**Limitations:**

yes.

**Strengths And Weaknesses:**

**Strenghts**

1.The motivation is clear and the problem definition is well-formulated. The paper clearly identifies the issue of geometric integrity destruction caused by existing spatial-domain mixup methods on point cloud data, and visualizes the discontinuous artifacts generated by spatial mixing.

2.The method design is theoretically grounded. The choice of using spherical harmonic transform (SHT) instead of graph Fourier transform is well justified: SHT provides a universal canonical basis, allowing the spectral coefficients of different point clouds to be directly combined in a mathematically valid linear fashion.

3.The experiments are comprehensive and convincing. Thorough evaluations are conducted on two standard benchmarks, covering various backbones (DGCNN, PointNeXt, Point-MAE) and 15 corruption types, and the results demonstrate the superiority of the proposed method.

4.The ablation study is well-designed. Three core components—randomized maximum degree of SHT, hierarchical mixing strategy, and adversarial rotation optimization—are separately validated, confirming the contribution of each component.

**Weaknesses**
1. All experiments are compared to works due to 2024, please add more results with baselines from 2025.

2. Authors should add an ablation study on the update frequency of the Adversarial Rotation Optimization, which is currently fixed to every five epochs. Experimenting with less frequent updates, such as every 10 or 20 epochs, could show whether the computational burden can be significantly reduced without sacrificing the reported 77.71 percent mean accuracy.

3. The choice of rotation initial value R_0 and other hyperparameters should be made more clear in the paper, considering this is a big part of reproducing

---

> ### Author Rebuttal · Authors · 2026-03-31
>
> We sincerely thank the reviewer for their thoughtful and constructive feedback. Below, we address each concern in detail:
>
> ---
>
> **W1: Additional Comparison with a Recent Baseline.**
>
> To address this concern, we add a new comparison with the recent SinPoint[1] and further conduct experiments on the PointMamba backbone. Results are summarized below:
>
> |Method|Avg.|
> |:---:|:---:|
> |Vanilla|60.59|
> |PointMixup|69.72|
> |RSMix|61.09|
> |PointCutMix-K|63.92|
> |PointCutMix-R|73.00|
> |SinPoint|59.16|
> |**PSMix-S**|**74.61**|
>
> From these experiments, our PSMix-S method achieves the highest mean accuracy ($74.61\%$), surpassing the best spatial-domain competitor (PointCutMix-R, $73.00\%$) by $1.61\%$. The results show that our method consistently improves robustness on stronger modern backbones, further supporting its general applicability as a plug-in augmentation framework.
>
> Notably, when directly running the released **SinPoint** code under **PointMamba + corruption robustness** setting, we obtain **$59.16$** average accuracy, which is below the **vanilla** baseline (**$60.59$**), although it achieves **$92.38$** on the corresponding **clean** test set. This suggests that SinPoint may be more aligned with improving clean-data generalization than corruption robustness. We will properly cite SinPoint and discuss this distinction in the revised paper.
>
> ---
>
> **W2: Ablation on the Update Frequency of Adversarial Rotation Optimization.**
>
> To empirically justify our selection (every 5 epochs), we performed a thorough ablation study using the PointNeXt backbone on the ModelNet40-C dataset, comparing several different update intervals. Results are presented below:
>
> |Setting|Density|Noise|Overall Trans|ROT|Mean(%)|
> |:---|:---:|:---:|:---:|:---:|:---:|
> |w/o|68.03|83.62|84.37|72.73|78.68|
> |update3|**68.29**|83.97|84.54|73.38|78.93|
> |update5|67.82|**84.78**|**84.71**|**73.50**|**79.10**|
> |update10|66.92|82.17|83.25|69.45|77.45|
>
> When the update frequency is reduced to every 3 iterations, PSMix-S achieves an average accuracy of $78.93\%$, which still demonstrates strong performance, with only a $0.17\%$ drop compared to the setting of 5. However, this higher frequency significantly increases computational cost by requiring more frequent adversarial rotations. Conversely, lowering the update frequency to every 10 iterations substantially reduces mean accuracy to $77.45\%$ (a drop of $1.65\%$), with a particularly severe decline of $4.05\%$ on rotation corruptions. This degradation likely arises from adversarial examples becoming outdated and less challenging, allowing the model to overfit. Therefore, updating the adversarial rotation optimization every 5 iterations provides the most effective balance between model robustness and computational efficiency.
>
> ---
>
> **W3: Clarification of $R_0$ Initialization and Hyperparameter Settings.**
>
> We agree that the initialization of $R_0$ and the hyperparameter settings should be stated more clearly for reproducibility. As described in the implementation details of our paper, we use an alternating optimization schedule initialized with **random Euler angles**, where the rotation parameters are updated every **5 epochs** by maximizing the classification loss in Eq. (15). During backbone updates, rotations are sampled from the current Euler parameters, with the sampling variance $\delta$ linearly annealed from $\pi$ to $\pi/36$. In addition, the maximum SHT degree $L$ is uniformly sampled from $[15,70]$, the transition threshold is fixed at $\tau = 3$, and all models are trained for $300$ epochs with batch size $32$ using Adam with an initial learning rate of $10^{-3}$. We agree that these details are currently somewhat scattered, and we will consolidate them more explicitly in the revised paper.
>
> ---
>
> [1] *Rethinking Point Cloud Data Augmentation: Topologically Consistent Deformation. ICML 2025.*

---

> > ### Author Rebuttal · Reviewer_Y442 · 2026-04-03
> >
> > Thanks for the reviewers' response. I believe that all three issues I raised have been well addressed, especially considering the authors' ablation experiments on paper parameters and additional comparative experiments, which significantly enrich the paper's practicality. Taking into account the paper's innovation and content comprehensively, I think the current score is quite reasonable.

---

> > > ### Author Response · Authors · 2026-04-03
> > >
> > > Thank you very much for your positive feedback and thoughtful reassessment. We sincerely appreciate your recognition of our revisions and additional experiments.

---

### Official Review · Reviewer_26od · 2026-03-14

**Soundness:** 3
**Presentation:** 3
**Significance:** 3
**Originality:** 3
**Overall Recommendation:** 4
**Confidence:** 3

**Summary:**

This paper proposes PSMix, a spectral-domain data augmentation method for point cloud classification. Instead of mixing point clouds directly in Euclidean space, the method first projects them onto the unit sphere and applies a spherical harmonic transform (SHT) to obtain frequency coefficients. Mixing is then performed hierarchically in the spectral domain, assigning different weights to low-frequency components (capturing global structure) and high-frequency components (capturing fine geometric details). Rotations are applied directly to the spectral coefficients using Wigner-D matrices, enabling rotation-equivariant and adversarial augmentation. The authors further introduce a spectral-ratio label mixing strategy to match the frequency-dependent interpolation. Experiments show improved robustness compared to spatial-domain Mixup baselines.

**Compliance With Llm Reviewing Policy:**

Affirmed.

**Final Justification:**

Update:

Thanks to the authors for their detailed clarification. The authors have provided analyses addressing each of my identified weaknesses and questions, and adequately addressed my concerns in the rebuttal. In particular, the response to Q1 is clear, and the additional theoretical analysis and experimental results provided for Q2 significantly strengthen the paper.

Based on these clarifications, I am satisfied with the revisions and will raise my score to 4 (Weak accept).

**Key Questions For Authors:**

1. What is the source of the performance gap compared to prior ModelNet40-C benchmarks — differences in experimental settings or implementation issues? Will the authors release the code to ensure reproducibility? If these concerns can be adequately addressed, I would be highly likely to increase my score.
2. Can the authors provide additional experiments or theoretical analysis to more clearly justify the semantic meaning of low- and high-frequency components in the spectral domain?
3. How does this work fundamentally differ from prior spectral-domain approaches beyond applying existing SHT and Wigner-D tools to data augmentation?

**Limitations:**

The paper has several limitations: (1) the experimental results show inconsistencies with established benchmarks, raising concerns about evaluation reliability; (2) the analysis of low- and high-frequency components is primarily qualitative and lacks rigorous validation; (3) the method is evaluated only on object-level classification and has not been demonstrated to generalize to more complex scene-level tasks; and (4) experiments are conducted on a limited set of backbones, without validation on more recent architectures.

**Strengths And Weaknesses:**

Strengths:
1. By transforming point clouds into the spherical harmonic domain, this design better preserves global geometric structure and avoids discontinuities or physically implausible artifacts introduced by direct coordinate interpolation.

2. The paper proposes a hierarchical spectral mixing scheme accordingly. The corresponding spectral-ratio label mixing and adversarial rotation optimization further demonstrate a thoughtful and comprehensive use of augmentation mechanisms.

3. Despite operating in the spectral domain, the approach maintains relatively low computational complexity under practical settings, making it feasible to integrate as a plug-in augmentation module without prohibitive training cost.

Weakness:

Major Issues:

1. The reported baseline results appear inconsistent with prior benchmark findings. For example, in the ModelNet40-C[1] benchmark paper, DGCNN achieves an error rate of 25.9% (≈74.1% accuracy) under a similar setting (300 epochs, 1024 points), whereas this paper reports only 66.76% for the Vanilla DGCNN baseline (in table1). Similar discrepancies are observed for PointCutMix and RSMix. This raises concerns that the baselines may not be fairly reproduced, and thus the reported improvements might be partially due to weaker reference implementations rather than intrinsic advantages of the proposed method.

[1] Benchmarking Robustness of 3D Point Cloud RecognitionAgainst Common Corruptions arXiv preprint arXiv:2201.12296

2. The experiments are conducted only on relatively older backbones such as DGCNN and PointNeXt. More recent and competitive models (e.g., RisurConv[1], Point Cloud Mamba[2], or other modern transformer/state-space architectures) are not evaluated. Given that the method is proposed as a general augmentation framework, validation on stronger and more up-to-date models would be necessary to convincingly demonstrate its broad applicability and practical impact.

[1] Risurconv: Rotation invariant surface attention-augmented convolutions for 3d point cloud classification and segmentation ECCV 202；

[2] Point Cloud Mamba: Point Cloud Learning via State Space Model AAAI 2025

3. The main technical building blocks (canonical spectral basis + Wigner-D rotation) are not new. The novelty therefore primarily lies in combining these tools with a Mixup-style augmentation framework. It would be helpful to more clearly articulate what is fundamentally new beyond applying existing spectral tools to data augmentation.

4. The claim that low frequencies encode global structure and high frequencies capture fine details is mainly supported by visualizations, which may be influenced by reconstruction resolution (smaller L naturally produces smoother shapes, while larger L increases detail). This does not conclusively justify the structural interpretation. A more convincing analysis would reconstruct shapes using only low- or high-frequency components and evaluate their semantic or geometric properties quantitatively.

5. The performance depends on the transition bandwidth τ (Figure 4). While the paper shows a sensitivity analysis, the optimal τ varies between datasets (5 for ModelNet, 3 for ScanObjectNN). Guidelines for selecting τ without extensive tuning would strengthen the practicality of the method.

6. The authors acknowledge in Section 5 that the method is currently focused on object-level classification. Direct application to large-scale, unbounded scenes (e.g., semantic segmentation) is non-trivial. Expanding on potential strategies for block-wise spectral processing would add value.

Minor Issues:

The spectral-ratio label mixing is derived by averaging frequency-dependent mixing coefficients, but the connection between frequency bands and semantic dominance could be more rigorously justified. A clearer theoretical explanation would improve clarity.

---

> ### Author Rebuttal · Authors · 2026-03-31
>
> We thank the reviewer for the thorough assessment. Below we address each concern.
>
> ---
>
> **W1&Q1&L1: Experimental Reliability and Reproducibility.**
>
> The performance gap stems from **different evaluation protocols**. ModelNet40-C reports results **averaged over severity 1–5**, while we report under **severity = 5 only**, the hardest level, naturally yielding lower accuracy. This stricter protocol applies equally to all methods. All baselines use their **officially released code and default hyperparameters** within the official ModelNet40-C repository. We will **release code upon acceptance**.
>
> ---
>
> **W2&L4: Evaluation on Advanced Models.**
>
> Our paper already includes **Point-MAE** (Transformer-based). We further evaluate on **PointMamba**. Due to space constraints, please refer to our response to **Reviewer Y442 (W1)** for a detailed discussion.
>
> ---
>
> **W3&Q3: Novelty Relative to Prior Work.**
>
> Prior spectral methods (HSP [1], GSDA [2], Texture [3]) rely on GFT, whose eigenbasis is derived from each sample's graph Laplacian. Since different point clouds yield different graphs, their eigenbases are **not aligned**, making cross-sample coefficient mixing mathematically ill-posed. Our core contribution is adopting SHT as a **universal, input-independent** basis, enabling principled cross-sample spectral mixing for the first time. This is not merely replacing GFT with SHT — we design a full pipeline around SHT's unique properties: (1) **Wigner D-matrix rotation** for differentiable, gimbal-lock-free spectral rotation; (2) **hierarchical frequency-adaptive mixing** leveraging the multi-resolution structure of spherical harmonics; and (3) **adversarial rotation optimization** via a minimax objective. Each component is enabled by the specific mathematical properties of SHT and together constitute a novel augmentation framework that goes well beyond prior spectral analyses of point clouds.
>
> ---
>
> **W4&Q2&L2: Semantic Interpretation of Frequency Components.**
>
> This interpretation is grounded in standard spectral analysis, not solely Fig. 3. Spherical harmonics $Y_l^m$ satisfy the Laplace–Beltrami eigenequation $-\Delta_{S^2}Y_l^m=l(l+1)Y_l^m$, where degree $l$ determines angular frequency: smaller $l$ captures smooth, global structure; larger $l$ captures fine-grained details. This is well-established in spectral geometry [4–5]. Fig. 3 is an intuitive visualization of this mathematically grounded property. With total reconstruction degree $l=60$, we define low frequency as $l\leq16$ and high frequency as $l>16$. On ModelNet40, keeping only low frequencies still gives 83.31% accuracy, while keeping only high frequencies drops to 15.28% (clean: 91.94%), supporting that low frequencies preserve global semantic structure and high frequencies mainly encode fine details.
>
> ---
>
> **W5: Practical Selection of $\tau$.**
>
> Fig. 4 shows consistently strong performance within $\tau \in [3,5]$, requiring only a small search over this range without extensive tuning.
>
> ---
>
> **W6&L3: Extension to Large-Scale Scenes.**
>
> We agree that applying global SHT to large-scale scenes is non-trivial, likely requiring **block-wise spectral processing**. To explore this, we implemented **PSMix-P**, which partitions each point cloud into **8 local patches** (256 points each), applies SHT per patch, performs spectral rotation and hierarchical mixing on local coefficients. Results on **ScanObjectNN-C** with **PointNeXt**:
>
> |Setting|Density|Noise|Trans|Mean|
> |---|---:|---:|---:|---:|
> |PSMix-S|50.50|46.69|**75.25**|57.47|
> |PSMix-P|**50.81**|**53.63**|72.77|**59.07**|
>
> PSMix-P substantially improves Density (+0.31%) and Noise (+6.94%), raising overall mean from 57.47% to 59.07%, while Transformation drops slightly (−2.48%). This suggests that patch-wise spectral processing strengthens robustness to local perturbations but weakens global geometric transformation modeling. These results indicate that block-wise spectral processing is a viable path for extending PSMix beyond object-level tasks.
>
> ---
> **Minor Issue: Semantic Meaning of Frequency Bands.**
> Our spectral-ratio label mixing is a practical approximation, not a strict semantic decomposition. Its motivation is that the mixed shape is constructed frequency band by frequency band, so the final semantic dominance should depend on the relative contribution of the two source shapes across bands. We agree that this connection can be explained more clearly, and will make the theoretical motivation more explicit in the revision.
>
> ---
>
> [1] *Hypergraph Spectral Analysis and Processing in 3D Point Cloud.* TIP 2021.
>
> [2] *Exploring the Devil in Graph Spectral Domain for 3D Point Cloud Attacks.* ECCV 2020.
>
> [3] *Texture: Text-guided Texturing of 3D Shapes.* ACM SIGGRAPH 2023.
>
> [4] *Learning the Spherical Harmonic Features for 3-D Face Recognition.* TIP 2012.
>
> [5] *Spectrum AUC Difference (SAUCD): Human-aligned 3D Shape Evaluation.* CVPR 2024.

---

> > ### Author Rebuttal · Reviewer_26od · 2026-04-01
> >
> > Thanks to the authors for their detailed clarification. The authors have provided analyses addressing each of my identified weaknesses and questions, and adequately addressed my concerns in the rebuttal. In particular, the response to Q1 is clear, and the additional theoretical analysis and experimental results provided for Q2 significantly strengthen the paper.
> >
> > Based on these clarifications, I am satisfied with the revisions and will raise my score to 4 (Weak accept).

---

> > > ### Author Response · Authors · 2026-04-01
> > >
> > > Thank you again for your positive feedback and for your willingness to raise the score. We sincerely appreciate your careful reading and constructive comments, which have helped strengthen the paper. We just wanted to kindly follow up, as the updated score does not seem to be visible yet on our side.

---

### Decision · Program_Chairs · 2026-04-30

**Decision:**

Accept (regular)

**Comment:**

This paper introduces mixing point clouds in the spectral domain, enabling hierarchical mixing from low to high frequencies. The reviewer appreciated the theoretical validity, the preservation of global topology, and the plug-in idea. Regarding concerns on the ablation study, lower accuracy than baselines, and limited to object-level classification, the authors provided extensive feedback. As a result, all reviewers clarified that the concerns were properly addressed and raised the scores to the positive side. AC agrees with the reviewers' recommendations and confirms that the paper has merit for publication.